# Satellite altimetry detection of ice shelf-influenced fast ice

Gemma. M. Brett[1], Daniel Price[1], Wolfgang Rack[1], and Patricia J. Langhorne[2]

[1]Gateway Antarctica, University of Canterbury, Christchurch, New Zealand
[2]Department of Physics, University of Otago, Dunedin, New Zealand

*Correspondence to*: Gemma M. Brett (gemma.brett@canterbury.ac.nz)

**Abstract.** The outflow of supercooled Ice Shelf Water from the conjoined Ross and McMurdo ice shelf cavity augments fast ice thickness and forms a thick sub-ice platelet layer in McMurdo Sound. Here, we investigate whether the CryoSat-2 satellite
radar altimeter can consistently detect the higher freeboard caused by the thicker fast ice combined with the buoyant forcing of a sub-ice platelet layer beneath. Freeboards obtained from CryoSat-2 were compared with four years of drill hole measured sea ice freeboard, snow depth, and sea ice and sub-ice platelet layer thicknesses in McMurdo Sound in November 2011, 2013, 2017 and 2018. The spatial distribution of higher CryoSat-2 freeboard concurred with the distributions of thicker ice shelf-influenced fast ice and the sub-ice platelet layer. The mean CryoSat-2 freeboard was 0.07-0.09 m higher over the main path of
supercooled Ice Shelf Water outflow, in the centre of the sound, relative to the west and east. In this central region, the mean CryoSat-2 derived ice thickness was 35 % larger than the mean drill hole measured fast ice thickness. We attribute this overestimate in satellite altimeter obtained ice thickness to the additional buoyant forcing of the sub-ice platelet layer which had a mean thickness of 3.90 m in the centre. We demonstrate the capability of CryoSat-2 to detect higher Ice Shelf Water influenced fast ice freeboard in McMurdo Sound. Further development of this method could provide a tool to identify regions
of ice shelf-influenced fast ice elsewhere on the Antarctic coastline with adequate information on the snow layer.

## 1 Introduction

Ice shelves are the floating extension of the grounded Antarctic ice sheet and buttress the flow of the grounded ice streams (Fürst
et al., 2016). Ice shelves and outlet glaciers comprise 74 % of the Antarctic coastline (Bindschadler et al., 2011) presenting an enormous interface where the ocean can directly interact with the grounded ice sheet. Ocean-driven basal melting near the grounding zones of outlet glaciers and the frontal zones of large ice shelves drives half of the net mass loss of ice shelves (Rignot et al., 2013; Depoorter et al., 2013). Cold and relatively fresh meltwater from ice shelves reduces the temperature and the density of the upper surface ocean, stabilising the upper water column and enhancing the thickness of sea ice near ice shelves (Hellmer,
2004; Gough et al., 2012; Purdie et al., 2006).

Fast ice is attached to the coast via land, ice shelves, glacier tongues or between shoals or grounded icebergs (Massom et al., 2010). When fast ice attaches to ice shelves or outlet glaciers it forms an important interface between the ice sheet and open ocean/pack ice (Giles et al., 2008; Massom et al., 2018). Fast ice affects ice sheet mass balance by providing mechanical stability and by buttressing glacier tongues (Massom et al., 2010) and ice shelves from the impacts of ocean swell (Massom et
al., 2018).

Coastal polynyas play a critical role in transporting heat energy from the ocean surface to the Antarctic ice sheet margin in the grounding zones of ice shelves and outlet glaciers (Silvano et al., 2018). Sea ice formation and brine rejection within coastal polynyas form a highly saline and dense water mass called High Salinity Shelf Water (HSSW) (Ohshima et al., 2016). HSSW
is maintained at the surface freezing temperature (~-1.9 °C) (Foldvik et al., 2004) and is sufficiently dense to circulate at depth into the cavities of adjacent outlet glaciers and ice shelves where it can drive basal melting in the grounding zone (Jacobs et

al., 1992) forming Ice Shelf Water (ISW) (Macayeal, 1984). ISW is characterised as being potentially supercooled (i.e., potential temperature below the surface freezing point) (Jacobs et al., 1985) and can rise in buoyant plumes along the base of ice shelves (Jenkins and Bombosch, 1995).


As ISW rises from depth, it can become in situ supercooled (Foldvik and Kvinge, 1974; Jenkins and Bombosch, 1995) (henceforth referred to as 'supercooled' for brevity) and frazil ice crystals can form (Holland and Feltham, 2005). The frazil ice can grow into larger platelet ice crystals (Langhorne et al., 2015; Leonard et al., 2011; Smith et al., 2001) which can freeze into the base of nearby sea ice to form consolidated platelet ice (Smith et al., 2012; Smith et al., 2001). The consolidation of

platelet ice augments sea ice formation (Eicken and Lange, 1989) and increases the sea ice thickness (Gough et al., 2012; Purdie et al., 2006). An unconsolidated mass of platelet ice crystals called a sub-ice platelet layer (SPL) can form beneath the sea ice once the conductive heat flux from the sea ice to the atmosphere becomes sufficiently low or the supply of frazil and platelet ice crystals abundant enough to overtake thermodynamic sea ice growth (Dempsey et al., 2010; Hoppmann et al., 2015; Gough et al., 2012). The signature of in situ supercooled ISW can thus be identified from thicker sea ice with incorporated

platelet ice and by the presence of an unconsolidated SPL (Langhorne et al., 2015).

In addition to the thicker ice shelf-influenced sea ice, the buoyant forcing of the SPL increases the sea ice freeboard (i.e., the height of the sea ice surface above sea level) (Gough et al., 2012; Price et al., 2014). Sea ice thicknesses obtained through satellite altimetry assumed to represent consolidated sea ice thickness are consequently overestimated (Price et al., 2014). The

magnitude of the SPL buoyancy is dependent on the thickness of the layer and the solid ice fraction, i.e., the fraction of solid ice per unit volume (Price et al., 2014). The buoyant forcing of a SPL with a thickness of one metre and a solid fraction of 0.25 has the potential to induce a 1-2 cm increase in freeboard height for typical first-year fast ice in McMurdo Sound (Gough et al., 2012).

The conditions of ice shelf geometry and sub-ice shelf circulation required for ISW to reach the upper surface ocean are not satisfied at all ice shelves (Langhorne et al., 2015) and the distribution of ice shelf meltwater in the Southern Ocean is not well known (Kusahara and Hasumi, 2014). Langhorne et al. (2015) collated observations of frazil and platelet ice in the upper surface ocean in Antarctic coastal regions and found positive occurrences where the water temperature at 200 m depth was less than 0.5°C above the surface freezing point (refer to Fig. 1a for a map of locations updated by Hoppmann et al. (2020)).

In the western Ross Sea, an interactive system is at play between coastal polynyas, the conjoined McMurdo and Ross Ice Shelf, and fast ice in the field study area in McMurdo Sound (Brett et al., 2020). ISW reaches the upper surface ocean in McMurdo Sound (Hughes et al., 2014; Lewis and Perkin, 1985; Mahoney et al., 2011; Robinson et al., 2014) where it significantly influences fast ice formation and forms a thick SPL.

The outflow of supercooled ISW from the McMurdo Ice Shelf cavity augments fast ice formation (Robinson et al., 2014), fast ice thickness (Gough et al., 2012; Langhorne et al., 2015; Leonard et al., 2006) and forms a thick SPL in McMurdo Sound (Dempsey et al., 2010; Gough et al., 2012; Langhorne et al., 2015). In the central-western region of the sound, a consistent yearly pattern of thicker ice shelf-influenced fast ice with a substantial SPL (Fig. 1b) beneath has been observed in proximity to the ice shelf margin in multiple studies, e.g., Brett et al. (2020); Price et al. (2014). This pattern is driven by the outflow of

supercooled ISW from the centre and west of the McMurdo Ice Shelf cavity and its subsequent circulation along the Victoria Land Coastline (Hughes et al., 2014; Lewis and Perkin, 1985; Robinson et al., 2014). The effect of supercooled ISW is most pronounced within ~30 km of McMurdo Ice Shelf (Brett et al., 2020; Hughes et al., 2014) but could extend up to 200-250 km north (Stevens et al., 2009; Hughes et al., 2014). ISW circulation has been modelled to increase fast ice growth by $9 \pm 4$ cm yr$^{-1}$ 100 km north of the ice shelf edge (Hughes et al., 2014). Indeed, thicker ice shelf-influenced fast ice (>2 m) and SPLs (0.1–

0.2 m) were measured in drill holes ~85 km north of the McMurdo Ice Shelf in 2013, 2016, and 2017 (Brett et al., 2020; Price et al., 2014). Brett et al. (2020) demonstrated a correlation of SPL thickness and volume in McMurdo Sound in late spring with a higher frequency of strong southerly wind events in the western Ross Sea which drive polynya activity, HSSW production and ISW formation and circulation within the McMurdo-Ross ice shelf cavity over winter.

Given the difficulty of accessing fast ice in Antarctic coastal regions and the logistical constraints of carrying out field observations with ground-based methods, it is possible that other unobserved regions are influenced by the outflow of ISW in the upper surface ocean. A means to identify these regions in large-scale satellite assessments is highly desirable, and if effective, has the potential to provide a satellite-based method to monitor the interactive system at play between the atmosphere, coastal polynyas, and circulation within ice shelf cavities with further development. The detection of ISW

influence on fast ice via satellite altimetry is in theory possible through the identification of regions with higher freeboard driven by thicker ice shelf-influenced sea ice, and if present, the buoyant forcing of a SPL (Price et al., 2014) but to date has not been assessed.

       The pulses emitted from satellite laser altimeters reflect from the ice-air or snow-air interface and thus measure the height of

the sea ice freeboard plus the addition of a snow layer, if present (henceforth referred to as snow freeboard). Microwave radar waveforms emitted from satellite radar altimeters penetrate into the snow layer. The penetration depth of radar waves into the snow layer is dependent on the backscattering properties of the snow (Kwok, 2014; Price et al., 2015; Willatt et al., 2011). The dominant backscattering surface for radar waveforms can thus be some unconstrained interface between the top of the snow and the sea ice freeboard (Price et al., 2015; Price et al., 2019).


       Satellite altimetry assessments of fast ice freeboard have previously been carried out in McMurdo Sound with the ICESat-1 laser altimeter (Price et al., 2013) and the CryoSat-2 (CS2) radar altimeter (Price et al., 2015; Price et al., 2019). In the ICESat-1 study, a peak in multi-year snow freeboard, centred around longitude 165°E, was observed in the south of McMurdo Sound between 2003 and 2009, when ocean circulation in the region was altered by the passage of large tabular icebergs (Robinson and Williams, 2012). The peak in multi-year fast ice snow freeboard coincided with the thickest SPL and the main path of

ISW outflow. Substantial multi-year fast ice thicknesses (~10-55m) measured with ICESat-1 beside the Mertz Glacier Tongue was proposed to be augmented by supercooled ISW and basal accretion of marine ice (Massom et al., 2010).

       The SPL contribution to satellite altimeter derived fast ice thickness in McMurdo Sound was quantified using surface elevation

measurements obtained with a Global Navigation Satellite System (GNSS) rover (Price et al., 2014). GNSS surface elevation (calibrated to local sea level) is analogous to satellite altimeter measured snow freeboard and was thus a comparable means to assess the effects of the SPL on freeboard to thickness conversion assuming hydrostatic equilibrium. Fast ice thicknesses derived from sea ice freeboard in McMurdo Sound were overestimated on average by 12 % and up to 19 % primarily due to the buoyancy effect of the SPL. Price et al. (2015) developed the method applied in this study to assess CS2 fast ice freeboard

in McMurdo Sound in 2011 and 2013 and the relevance of this work to this study is described in more detail in sect. 2.2.

       For the first time, we investigate whether the CS2 satellite radar altimeter can detect the influence of ISW on fast ice in McMurdo Sound by consistently identifying the higher ice freeboard caused by thicker ice shelf-influenced fast ice and the buoyant forcing of the SPL beneath. Multiple years of field observations and a highly detailed knowledge of the spatial distributions of snow, fast ice and SPL thicknesses, and the circulation of the supercooled ISW recommends McMurdo Sound

as an ideal location for this study. CS2 measurements of freeboard obtained using a supervised retrieval procedure were compared with four years of drill hole measured freeboard, fast ice and SPL thickness, and snow depth over the fast ice in

McMurdo Sound. We assess and compare spatial trends in CS2 freeboard and CS2 ice thickness in a region with significant ISW influence (centre) and another region with less pronounced ISW influence (east) (Fig. 4). We describe the study area, in situ datasets and summarise the technical specifications of the CS2 satellite radar altimeter and CS2 data product in sect. 2. We describe the methods applied in sect. 3 and the results in sect. 4. In sect. 5, we discuss the results and provide an outlook for satellite altimetry assessments of ice shelf-influenced fast ice.

## 2    Study area and datasets

### 2.1 Ground validation with drill hole measurements

McMurdo Sound is geographically delineated by the Victoria Land Coastline in the west, Ross Island in the east, and the McMurdo Ice Shelf in the south (Fig. 1). The fast ice in McMurdo Sound is predominately smooth and undeformed first-year ice that typically forms between April and December and then breaks out in the following summer (Kim et al., 2018). Ground validation for the CS2 measured freeboard and derived ice thickness was provided by extensive drill hole measurements made at field sites distributed at ~5-10 km spacing on the fast ice in McMurdo Sound in November 2011, 2013, 2017 and 2018. At each site, five drill holes were made in the sea ice at the centre and end points of two 30 m cross-profile lines. Sea ice freeboard, snow depth and the thicknesses of sea ice and the SPL were measured at each drill hole using the technique described in Price et al., 2014 and then averaged to give a representative value over the 30 m by 30 m area.

In 2011and 2018, measurements were respectively made at 43 (Langhorne et al., 2021a) and 14 (Brett et al., 2021) sites distributed over an area of ~1500 km$^2$ in the south of McMurdo Sound. Fast ice in the northwest was additionally assessed in 2013 and 2017 with 34 (Rack et al., 2021) and 21 (Langhorne et al., 2021b) sites, respectively, distributed over an area of ~3000 km$^2$. Interpolated maps of snow depth, fast ice and SPL thicknesses and sea ice and snow freeboard at field sites were generated for each year. In multiple field seasons in McMurdo Sound, we observed smooth gradients in the thickness of sea ice, SPL and the snow layer. We thus applied a minimum curvature (i.e., thin plate) first derivative spline interpolation which passes through data points and no smoothing. Coincident interpolated values for each parameter were then extracted for each CS2 measurement point along-track. Interpolated drill hole snow and ice freeboard could not capture small-scale variability given that drill hole measurements were 1-10 km apart. However, it was expected that the interpolations would represent the smooth gradients in sea ice, SPL and snow thicknesses observed in McMurdo Sound well enough on the large-scale to facilitate a comparison with the CS2 footprint (~300 m along-track and ~1.5 km across-track).

Brett et al. (2020) provide a detailed description of the thickness distributions of ice shelf-influenced fast ice, the SPL and snow in McMurdo Sound in November of 2011, 2013 and 2017. Here, we summarise those descriptions to show general patterns and also include the fast ice conditions in 2018. The spatial distributions of fast ice and the SPL were characteristic of McMurdo Sound with thicker ice shelf-influenced fast ice and a very thick SPL near the McMurdo Ice Shelf in the south, in the main path of ISW outflow. The fast ice and SPL thinned to the east, northeast and more gradually to the northwest. The SPL distribution observed each year is exemplified in Fig. 1b with a spline interpolated map from drill hole measurements made in November 2013. Higher drill hole measured freeboard concurred with the distributions of thicker fast ice and SPL. The snow distribution was consistent each year with more substantial deposition in the east and southeast (~0.2-0.4 m), and a sparse dusting (0-0.10 m) in the centre and west where the influence of ISW is greatest. In November 2018, the fast ice consisted of a 7-15 km wide band of two year old multi-year ice that ran parallel to the McMurdo Ice Shelf in the south and first-year ice in the north. The SPL beneath the second-year fast ice in 2018 had a maximum thickness of 11 m near the ice shelf in the south.

In McMurdo Sound, the 'anomalously' higher sea ice freeboard is driven by the combined upward (positive) buoyancy of the thicker ice shelf-influenced fast ice and the ice mass within the SPL beneath. The addition of a snow layer, if present, provides an opposing downward (negative) forcing. The weight of the snow can depress sea ice freeboard and result in flooding of the sea ice surface and the formation of meteoric ice which can contribute to freeboard (Maksym and Markus, 2008). Snow-depressed negative freeboard or surface flooding were not observed at drill hole sites in McMurdo Sound in late spring. Multiple ice core studies carried out in the region over winter and in late spring revealed no contribution of meteoric ice to the fast ice cover in McMurdo Sound (e.g., Dempsey et al., 2010, Gough et al., 2012).

A combined 'Mass Equivalent Thickness' (MET) was calculated according to Price et al. (2019) to sum the consolidated sea ice thickness and the unconsolidated SPL thickness times the solid fraction, i.e., the thickness of the solid mass of ice in the SPL. It is difficult to physically measure the solid fraction without disturbing the natural state of the SPL. Numerous methods to indirectly derive the solid fraction of the SPL in McMurdo Sound have been applied including the hydrostatic equilibrium assumption (0.16) (Price et al., 2014) and thermistor probe data (0.25) (Gough et al., 2012) with a range values from 0.16 to 0.5 derived (Gough et al., 2012). We assumed an intermediary solid fraction value of 0.2 between that derived by Gough et al. 2012 and Price et al., 2014 to calculate MET from the interpolated drill hole measurements for each year.

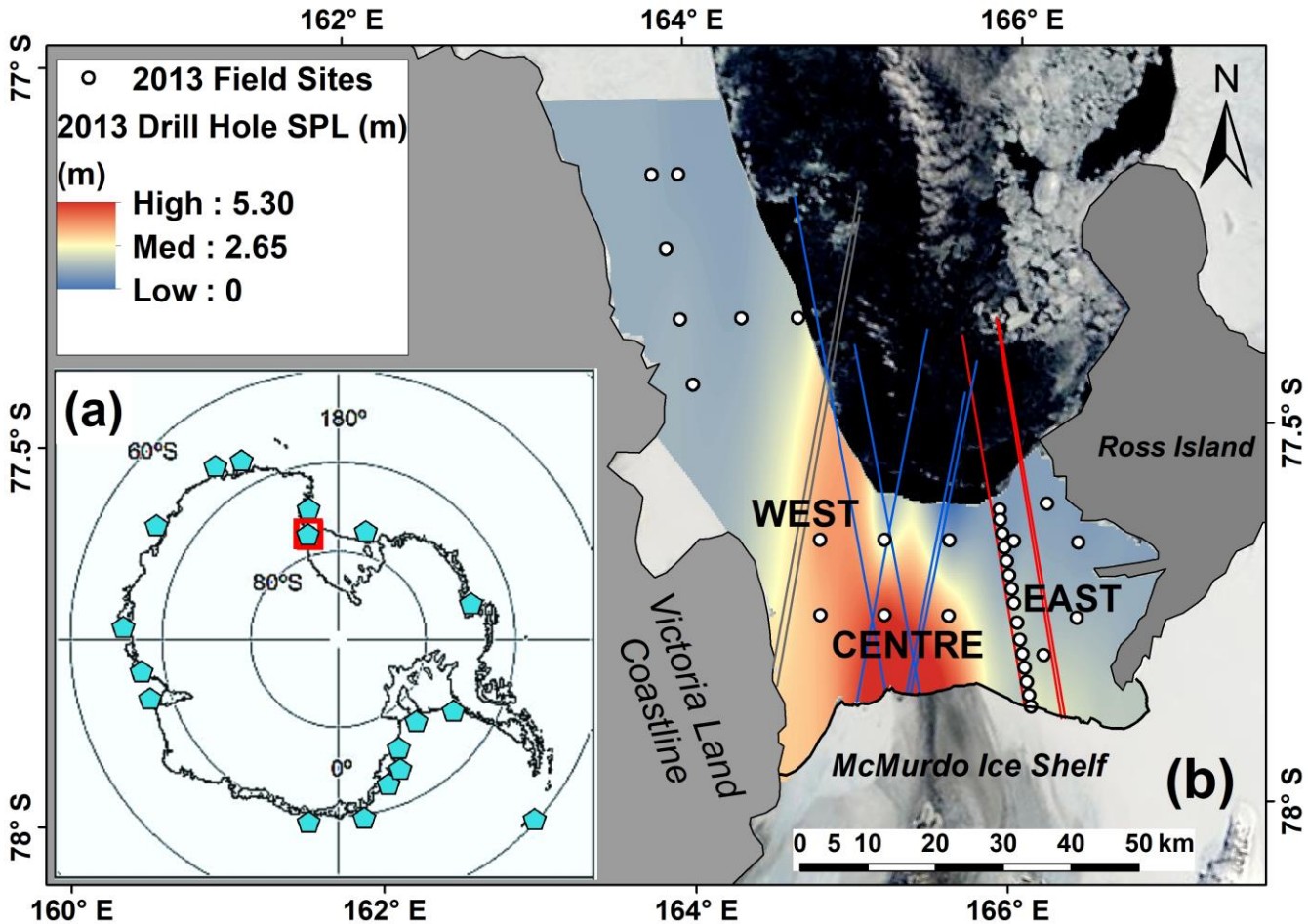

**Figure 1: (a) Antarctic inset map showing locations of positive occurrences of frazil or platelet ice (adapted from Langhorne et al. (2015) and updated by Hoppmann et al. (2020)) showing the location of (b) the McMurdo Sound study area (red square) in the western Ross Sea with CryoSat-2 tracks from multiple years included in the analysis in Table 1 and Fig. 3 and Fig. 4 for the west (3 tracks) (grey), centre (7 tracks) (blue) and east (4 tracks) (red) displayed on a spline interpolated map of drill hole measured SPL thickness in November 2013 (satellite image: NASA MODIS, 19 November 2013).**

## 2.2 CryoSat-2 satellite radar altimeter

The principle payload of CS2 is the combined Synthetic Aperture Radar (SAR) and interferometric radar altimeter which operates in three modes: 1) Low Resolution Mode, 2) SAR mode, and 3) Combined SAR and interferometer (SARIn) mode (Bouzinac, 2012; Wingham et al., 2006). CS2 predominantly operates in SARIn mode over coastal fast ice regions including McMurdo Sound. The number of measurements obtained in SARIn mode decreases by a factor of four relative to SAR mode (Wingham et al., 2006). Highly specular reflectors such as leads significantly off-nadir can dominate the returning radar echo or 'snag' introducing a range error and an underestimate in surface elevation. However, in SARIn mode this range error can be corrected by determining the across-track angle to off-nadir returns and more data can thus be retained relative to SAR mode (Armitage and Davidson, 2014). Depending on the surface geometry and satellite orbit, the radar footprint of CS2 is ~300 m along-track and ~1.5 km across-track (Wingham et al., 2006).

The ESA Level 2 retracking point is found using a model-fitting approach and is defined relative to the entire echo waveform (Bouzinac, 2012; Price et al., 2015). The dominant backscattering surface assumed for this retracker is the snow-ice interface. However, an assessment of the influence of the snow layer on the radar waveform by comparison with coincident in situ measurements of freeboard and snow layer depth in McMurdo Sound by Price et al. (2015) showed that the ESA retracker tracked between the surface of the snow and the snow-ice interface. Price et al. (2019) assessed the sensitivity of sea ice thicknesses derived from the CS2 Level 2 SARIn product to variable penetration depths into the snow layer in McMurdo Sound using the same in situ measurements from late spring November 2011. They found the closest agreement with in situ measurements when the penetration depth into the snow layer was assumed to be 0.05-0.10 m. In this study, we compare CS2 obtained freeboard with in situ measurements to identify the best matching freeboard interface or penetration depth for each individual CS2 track according to Price et al., 2019.

Geo-located Baseline-C Level 2 SARIn data generated from the 'CryoSat-2 Ice Processor' were used for this assessment. The Level 2 SARIn data product provides a surface height relative to the WGS84 ellipsoid for each location along-track with geophysical corrections applied to the range measured by the satellite (Wingham et al., 2006; Bouzinac, 2012). Applied geophysical corrections vary according to the surface type classification and are described in detail with their sources in Bouzinac (2012) and Webb and Hall (2016). Atmospheric propagation corrections (*ionospheric and dry/wet tropospheric*) are always applied to account for the time delay introduced as the altimeter pulse passes through the Earth's atmosphere. Satellite altimeter derived surface height must additionally be corrected for the shape of the geoid, tidal height, inverse barometric effect and dynamic ocean topography before freeboard can be obtained (Price et al., 2015; Ricker et al., 2016). However, the level of accuracy required for these corrections is difficult to attain with models. In satellite altimetry sea ice assessments, freeboard is generally determined relative to a local reference sea surface height (SSH) obtained over open water along the satellite track.

The fast ice region in McMurdo Sound is within the surface type mask of *open ocean* where ocean surface and tidal corrections are applied. Ocean surface corrections (*Sea State Bias* and *Dynamic Atmospheric*) account for the effect of the atmosphere on SSH. Tidal corrections (*Ocean Tides* and *Long-Period Equilibrium Tide*) are applied to adjust the range so that it appears to originate from the mean tide-free sea surface. Tidal-induced deformation of the Earth's crust is accounted with the *Ocean Loading, Solid Earth*, and *Geocentric Polar* corrections. The *Mean Sea Surface (MSS)* is a combination of the geoid and the mean dynamic topography and is accounted for over open ocean (Ricker et al., 2014; Skourup et al., 2017). To provide confidence in CS2 derived freeboard, an assessment of the spatial distribution and magnitude of the geophysical corrections applied to the CS2 Level 2 SARIn product and the effect of de-trending for the geoid using the Earth Gravitational Model 2008 (EGM 2008) (Pavlis et al., 2012) was carried out over ice-free open ocean in McMurdo Sound in late summer of 2011 to 2017. This is described in detail in Appendix A.

# 3 Method

## 3.1 Supervised CryoSat-2 retrieval of ISW influenced freeboard

By assessing individual CS2 track profiles, significant CS2 height outliers (i.e., WGS84 -54 m≤CS2 height≤-60 m) were identified and removed. Surface elevation was then obtained by applying the *MSS* and all ocean/tidal corrections and then de-

trending for the EGM 2008 geoid (refer to Appendix A for further information). A supervised retrieval procedure, according to Price et al., 2015, was applied to obtain freeboard from the Level 2 CS2 surface elevations by identifying the relative SSH manually with satellite imagery. The fast ice edge and open water along-track was identified in NASA MODerate resolution Imaging Spectroradiometer (MODIS) optical images acquired on the day of the CS2 overpass. The relative SSH was defined as the median value of the CS2 surface elevation retrievals over 25 km of adjacent open water along-track. The median value

was chosen because the mean was skewed by significant noise in the CS2 measurement. This distance provided ~80 CS2 surface elevation measurements to obtain a representative median value from the CS2 measurement and was also applied by Price et al, 2015 in their supervised freeboard retrieval. The distance on the fast ice to open water did not exceed ~25 km in all study years.

Supervised retrievals of fast ice freeboard were applied to 20 CS2 tracks between latitudes 77.4°S and 78°S in McMurdo Sound over the four study years. CS2 freeboards derived relative to the median SSH were then compared with coincident spline interpolated drill hole measured snow and ice freeboards along-track. Fourteen out of the 20 (70 %) tracks produced CS2 freeboard magnitudes relative to the median SSH that aligned with the drill hole measurements. The 14 CS2 tracks consisted of four in the east, 7 in the centre, and 3 in the west (refer to Fig. 1 for regional locations). Figure 2 shows an along-

track profile of spline interpolated drill hole measured sea ice and snow freeboard, and sea ice and SPL thicknesses and combined MET in the centre of the sound with CS2 freeboard measurements from an overpass on the 15 November 2017. CS2 freeboard increased towards the ice shelf in the south with increasing fast ice and SPL thicknesses and combined MET.

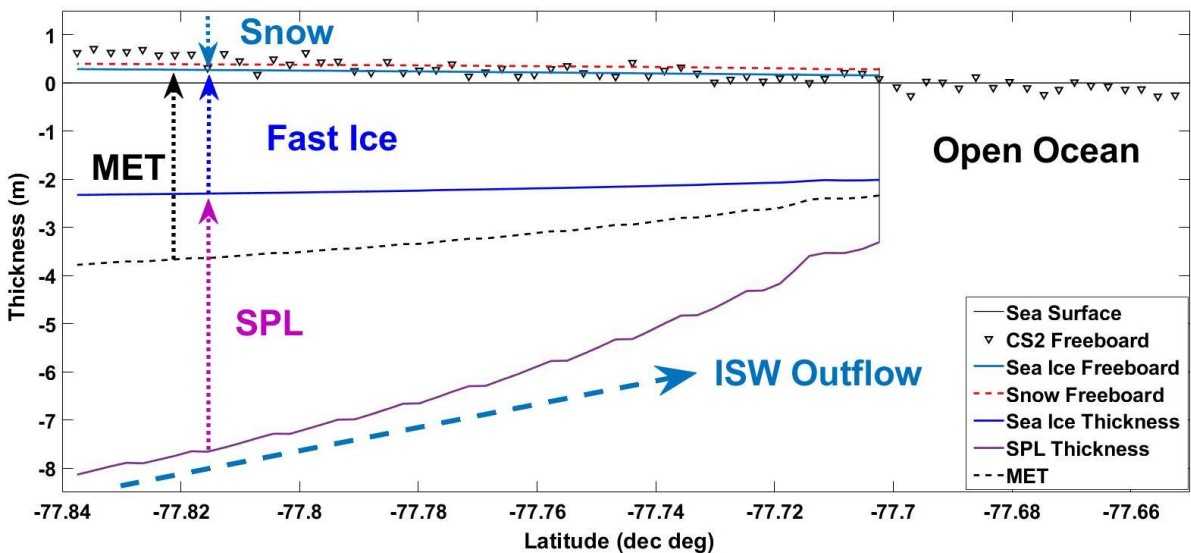

**Figure 2: Spline interpolated drill hole measured sea ice and snow freeboard, and fast ice, SPL and Mass Equivalent Thicknesses (MET) relative to the zero level reference sea surface from November 2017 with CS2 freeboard on the 15 November 2017. The McMurdo Ice Shelf is located in the south (left hand side) of this profile at approximately latitude -77.84°.**

Sea ice thickness ($T_i$) was calculated from freeboard assuming hydrostatic equilibrium which asserts that the ratio of freeboard

($Fb$) to total ice thickness is proportional to the ratio of the densities of sea ice ($\rho_i$) and seawater ($\rho_w$) (Zwally et al., 2008). The addition of the snow layer ($T_s$) to freeboard height, the snow density ($\rho_s$), and the buoyancy effect of the SPL, if present, must be taken into account (Price et al., 2014). Constant values for $\rho_w$, $\rho_i$ and $\rho_s$ of 1027 kg m$^{-3}$, 925 kg m$^{-3}$ and 385 kg m$^{-3}$, were

respectively assumed to facilitate inter-comparison with, and to adhere to the rationale and error propagation of Price et al., 2014 where a range of snow and sea ice density measurements made in McMurdo Sound were assessed.


CS2 ice thickness was calculated from CS2 freeboard by assuming that the dominant backscattering surface for the ESA Level 2 retracker is either 1) snow freeboard, 2) sea ice freeboard, or 3) some penetration factor ($Pf$) into the snow layer by respectively applying equations 1, 2 and 3 from Price et al. (2019). A correction for the propagation of the radar wave through the snow layer was applied according to Kurtz et al. (2014).


$$T_i = \left(\frac{\rho_w}{\rho_w - \rho_i}\right) Fb - \left(\frac{\rho_w - \rho_s}{\rho_w - \rho_i}\right) T_s \tag{1}$$

$$T_i = \left(\frac{\rho_w}{\rho_w - \rho_i}\right) Fb + \left(\frac{\rho_s}{\rho_w - \rho_i}\right) T_s \tag{2}$$

$$T_i = \left(\frac{\rho_w}{\rho_w - \rho_i}\right) Fb - \left(\frac{\rho_w - \rho_s}{\rho_w - \rho_i}\right) T_s + \left(\frac{\rho_w}{\rho_w - \rho_i}\right) Pf \tag{3}$$

To select the best-matching freeboard interface for each track, the profiles of CS2 ice thicknesses obtained from Eq. (1) and (2) were compared with interpolated drill hole METs along-track. The freeboard interface that produced the closest matching CS2 ice thickness and drill hole MET was then selected. This was assessed by 1) comparing along-track profiles of interpolated in situ MET with CS2 ice thicknesses visually (this was evident if the best interface was sea ice or snow freeboard), 2) linear regression analyses (if there was a gradient in freeboard with increasing fast ice and SPL thickness), or 3) comparing mean

values along-track according to Price et al., 2019. For two tracks in the east, neither snow freeboard (Eq. (1)) nor sea ice freeboard (Eq. (2)) produced aligning drill hole MET and CS2 ice thickness profiles, and thus variable penetration depths were applied in 0.01 m increments (Eq. (3)). The penetration factor which produced the smallest mean difference (summed along track) between the CS2 ice thickness and drill hole MET was selected according to Price et al., 2019.

The sea ice surface was the dominant freeboard interface in the west and centre, except for 2011 and 2017 which had deeper snow coverage across the sound. In the east, the best matching freeboard interfaces were the sea ice surface in 2017, the snow surface in 2013, and penetration depths of 0.11 m and 0.12 m into the snow layer in 2011 and 2018, respectively. If sea ice freeboard or a penetration depth was determined for a CS2 track, a correction for the propagation of the radar wave through the snow was applied according to Kurtz et al. (2014).



## 4 Results

### 4.1 Supervised CryoSat-2 retrieval of ISW influenced freeboard

The mean drill hole sea ice and snow freeboards, and snow-corrected CS2 freeboard for first-year fast ice were calculated (over equivalent distances) for each CS2 track, and then averaged regionally for the 3 tracks in the west, 7 in the centre and 4 in the east (refer to Fig. 3 and Table 1). The mean regional drill hole sea ice and snow freeboards were highest in the centre at 0.25 m and 0.32 m, respectively, and slightly lower in the west at 0.22 m and 0.27 m. In the east, the mean drill hole sea ice and snow freeboards were 0.14 m and 0.30 m, respectively. The mean regional CS2 freeboard followed the same trend of higher freeboard in the centre (0.31 m), decreasing to the west (0.24 m) and east (0.22 m). In the centre of McMurdo Sound, we estimate a mean CS2 derived freeboard difference of 0.07 and 0.09 m relative to the west and east, respectively. This difference is 28-36 % of the mean drill hole measured sea ice freeboard. The regional mean interpolated drill hole sea ice, SPL thicknesses and combined MET of the seven CS2 tracks in the centre of McMurdo Sound were 2.26 m, 3.90 m, and 3.08 m, respectively. The mean CS2 ice thickness for these seven tracks was 3.04 m, corresponding to a 0.78 m overestimate relative to the drill hole measured sea ice thickness in this region. The regional mean snow depth in the centre was 0.07 m over the four study years. We interpret the CS2 freeboard distribution across the main path of supercooled ISW outflow as being largely due to the thickness change of the ice shelf-influenced fast ice with a SPL beneath, rather than due to gradients in snow thickness.

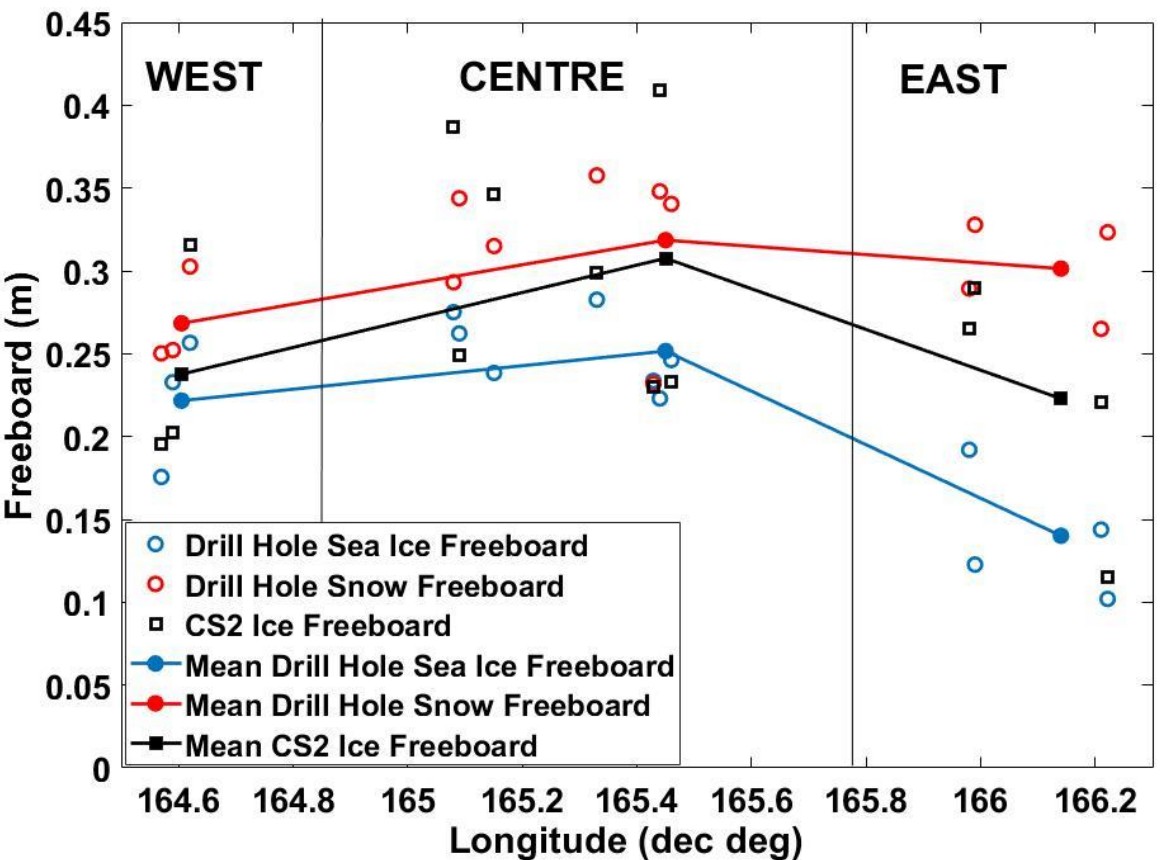

**Figure 3: Comparison of all drill hole measured sea ice and snow freeboards with CS2 freeboard over first-year fast ice only for the 3 tracks in the west, 7 in the centre and 4 in the east (refer to Fig. 1 for locations and values given in Table 1). The mean regional values for the tracks in the west, centre and east are shown with a solid line to illustrate the mean freeboard distribution across McMurdo Sound. The longitudes are taken from where each CS2 track crossed latitude 77.7°S.**

**Table 1: Mean interpolated drill hole (DH$_{int}$) measured sea ice and snow freeboards (Fb) and CS2 freeboards for first-year fast ice only (over equivalent distances) for individual CS2 tracks in the west, centre and east. Regional mean values for tracks in the west, centre and east are given and these statistics are used in Fig. 3 and Fig. 4. If sea ice freeboard or a penetration depth was determined for a CS2 track, a correction for the propagation of the radar wave through the snow was applied according to Kurtz et al. (2014).**

| CS2 Track ID (yyyymmdd) | (a) Mean DH$_{int}$ Sea Ice Fb (m) | (b) Mean DH$_{int}$ Snow Fb (m) | (c) Mean CS2 Fb (m) |
|---|---|---|---|
| *WEST* | | | |
| *20171117* | 0.18±0.01 | 0.25±0.01 | 0.20±0.23 |
| *20131105* | 0.23±0.00 | 0.25±0.01 | 0.20±0.14 |
| *20181120* | 0.26±0.00 | 0.30±0.00 | 0.32±0.14 |
| *Regional Mean* | **0.22** | **0.27** | **0.24** |
| *CENTRE* | | | |
| *20131129* | 0.28±0.03 | 0.29±0.00 | 0.39±0.11 |
| *20111123* | 0.26±0.03 | 0.34±0.00 | 0.25±0.12 |
| *20111127* | 0.24±0.03 | 0.32±0.03 | 0.35±0.12 |
| *20181115* | 0.28±0.01 | 0.36±0.02 | 0.30±0.16 |
| *20131103* | 0.23±0.05 | 0.23±0.05 | 0.23±0.11 |
| *20171115* | 0.22±0.04 | 0.35±0.03 | 0.41±0.18 |
| *20181118* | 0.25±0.14 | 0.34±0.14 | 0.23±0.12 |
| *Regional Mean* | **0.25** | **0.32** | **0.31** |
| *EAST* | | | |
| *20131127* | 0.19±0.04 | 0.29±0.10 | 0.27±0.11 |
| *20111121* | 0.12±0.04 | 0.33±0.02 | 0.29±0.21 |
| *20181113* | 0.14±0.01 | 0.27±0.04 | 0.22±0.13 |
| *20171110* | 0.10±0.03 | 0.32±0.03 | 0.12±013 |
| *Regional Mean* | **0.14** | **0.30** | **0.22** |

Along-track profiles of CS2 freeboard and CS2 ice thickness are compared in the centre and east with interpolated drill hole measurements for 2011, 2013, 2017 and 2018 in Fig. 4. Linear fits to CS2 ice thicknesses are shown with interpolated drill hole sea ice and SPL thicknesses, MET, and sea ice and snow freeboard. The linear fits were applied to CS2 ice thickness obtained over first-year ice only and did not include second-year ice. The linear fits applied to CS2 ice thickness show increasing trends towards the ice shelf in the centre for all years, in contrast to the east which had flat to marginal gradients in CS2 ice thickness.

Over the main supercooled ISW outflow region in the centre of the sound, CS2 freeboard and CS2 ice thickness increased towards the ice shelf in the south concurrently with increasing fast ice and SPL thicknesses and combined MET (Fig. 4a, c, e, and g). In comparison, CS2 freeboard and CS2 ice thickness in the east where the influence of ISW is less pronounced, showed a marginal increase towards the ice shelf in 2013 (Fig. 4d) and almost flat profiles in 2011, 2017 and 2018 (Fig. 4b, f, and h respectively). In 2011 (Fig. 4a) and 2017 (Fig. 4e), CS2 freeboard was higher near the McMurdo Ice Shelf in the centre of the sound and this could be attributed to the thicker SPLs or the deeper and more wind-compacted snow observed in these years by Brett et al. (2020). In 2011, no drill hole measurements were made south of latitude -77.83° on the centre profile (Fig. 4a) and we were unable to determine why the first-year fast ice freeboard was significantly higher near the McMurdo Ice Shelf. CS2 freeboard measured south of -77.83° on this track was thus excluded from the calculation of the mean value (Table 1).

In 2013, the CS2 track in the centre (Fig. 4c) on 3 November had minimal snow coverage and best matched with sea ice freeboard. The CS2 track in the east (Fig. 4d) on 27 November 2013 had deeper snow and best matched with snow freeboard. The drill hole measurements made every kilometre along the CS2 track in the east on 27 November 2013 by Price et al. (2015) are shown in this profile (Fig. 4d). The increasing trend towards the McMurdo Ice Shelf is evident in both the CS2 freeboard and CS2 ice thickness on the 2013 central profile when compared to the 2013 eastern profile, most markedly over second-year ice in the southwest of the sound (77.85°S to 77.87°S) in Fig. 4c.

To assess the spatial distributions of higher satellite altimeter obtained freeboard and resultant thicker ice, CS2 freeboard and CS2 ice thickness from 4 tracks distributed across McMurdo Sound in November 2013 were spline interpolated. To circumvent substantial noise in CS2 freeboard and derived ice thickness, a running mean of three measurements (corresponding to ~1 km) was applied along-track prior to applying the interpolation. Figure 5 shows maps of the distributions of CS2 freeboard, CS2 ice thickness and drill hole measured MET in McMurdo Sound in November 2013. CS2 freeboard height (Fig. 5a) and CS2 ice thickness (Fig. 5b) concurred with the thickness distributions of drill hole measured ice shelf-influenced fast ice, SPL (Fig. 1) and combined MET (Fig. 5c) with higher freeboards and ice thicknesses observed in the central-western region of the sound in the main path of supercooled ISW outflow. The trend of increasing CS2 freeboard and CS2 ice thickness towards the McMurdo Ice Shelf in the centre and west is evident.

370

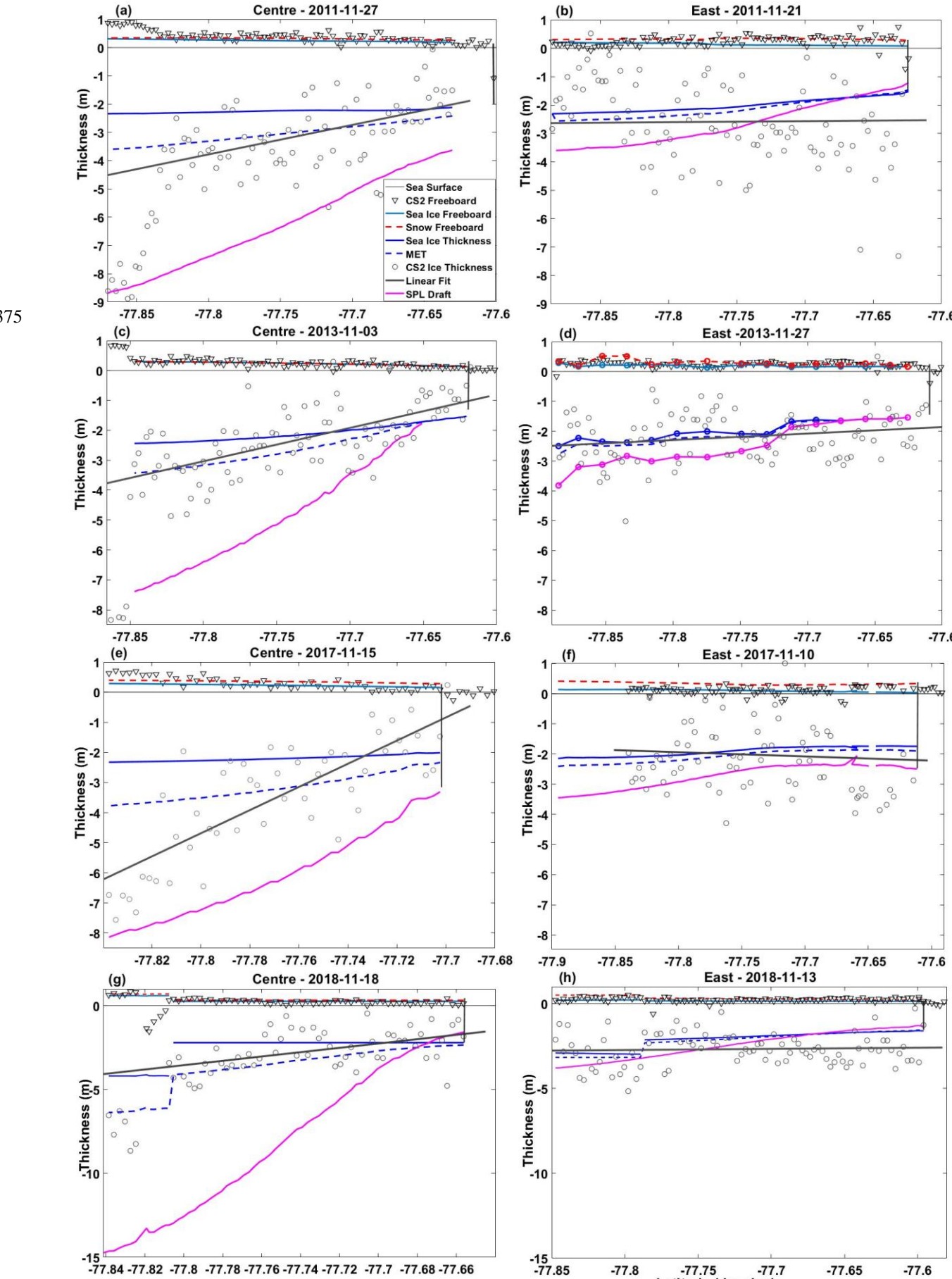

**Figure 4: Comparison of along-track profiles of CS2 freeboard and CS2 ice thickness in the centre (a, c, e, and g) and east (b, d, f and h) for 2011, 2013, 2017 and 2018. The McMurdo Ice Shelf is located in the south (left) of all profiles. Linear fits applied to CS2 ice thickness (over first-year ice only) are shown with interpolated drill hole sea ice and SPL thicknesses, MET and sea ice and snow freeboard. A drill hole validation line was carried on along the eastern CS2 track on the 27th November 2013 by Price et al. (2015) and drill hole measured snow and sea ice freeboard, sea ice and SPL thickness, and MET are included in (d) (circles).**

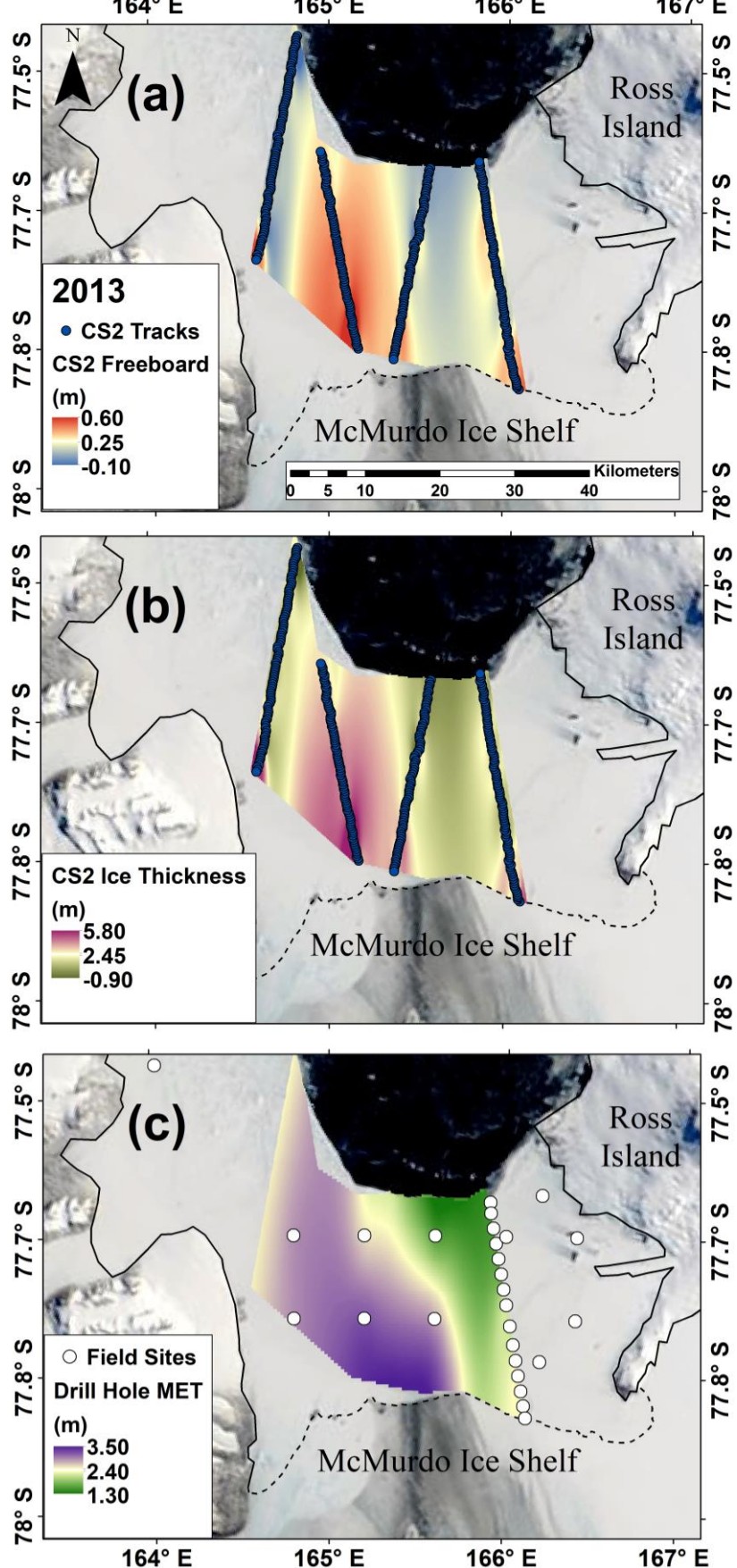

**Figure 5: CS2 freeboard was higher and CS2 ice thickness greater where supercooled ISW outflows in McMurdo Sound. Spline interpolated maps of the distributions of (a) CS2 freeboard, and (b) CS2 ice thickness from 4 tracks over first-year fast ice, and (c) interpolated drill hole MET measured at field sites in November 2013. A running mean of three CS2 freeboard and CS2 ice thickness measurements (corresponding to ~1 km) was applied along-track prior to applying the interpolation in a and b (satellite image: NASA MODIS, 19 November 2013).**

**5 Discussion**

**5.1 Geophysical corrections and identification of a relative sea surface height**

The proximity of fast ice to the coastline can introduce uncertainty in surface elevation retrievals from satellite altimetry. This
uncertainty is caused by increased complexity of the geoid near the continental landmass and by the interaction of the tides with the coastline and shallow bathymetry of the continental shelves. These factors are particularly relevant in McMurdo Sound. The mountainous terrain to the east and west and the ice shelf to the south affect the shape of the geoid. To ensure that the trends observed in CS2 freeboard did not result from inaccuracies in the geophysical corrections, an assessment of the spatial distribution and magnitude of the geophysical corrections applied to the CS2 Level 2 SARIn product and the effect of
de-trending for the geoid using the EGM 2008 was carried out over open ocean in late summer. Additionally, four years of in situ measurements over fast ice in late spring provided confidence in the CS2 Level 2 SARIn product and applied geophysical corrections. The remaining uncertainty in the CS2 freeboard captured in the standard deviations in Table 1 is likely driven by noise in the CS2 measurement and variability in the snow layer and the dominant backscattering interface along-track.

The geophysical corrections should not have a major impact on the obtained CS2 freeboard if the retrieval of the relative SSH is robust (Ricker et al., 2016). Identifying the relative SSH along-track is complicated by interference and noise introduced by sea surface conditions and by the presence of pack ice. We observed thin nilas or some pack ice beneath several CS2 tracks. However, sea surface conditions will have a more significant effect for automatic freeboard retrievals which interpret the backscatter or pulse peakiness of the returning signal to identify open water (Price et al. 2015). To ensure minimum error in
the relative SSH, we applied a supervised freeboard retrieval procedure where open water was manually identified in satellite imagery. The distance on the fast ice to open water did not exceed ~25 km in all study years. We used in situ measurements to assess the accuracy of the relative SSH identification by comparing the magnitude of the resultant CS2 freeboards against drill hole measured freeboard. Additionally, we assessed linear and spatial trends in CS2 freeboard and CS2 ice thickness in this study, e.g., Figs. 4 and 5. Any error in the relative sea surface height should have minimum effect on the trends because a
constant median SSH value is subtracted from CS2 surface elevation measurements to obtain CS2 freeboard for each individual track. However, for automated freeboard retrievals and for regions of coastal sea ice without in situ measurements or open water nearby, poor identification of the relative SSH could introduce significant error and bias in the CS2 derived freeboard.

**5.2 Satellite altimetry measured freeboard and sea ice thickness**

The assumed freeboard interface will significantly affect the resultant sea ice thickness obtained from the hydrostatic equilibrium equations. The calculated CS2 ice thickness will be overestimated if sea ice freeboard is assumed and full penetration of the radar wave into the snow layer does not occur. Alternatively, calculated CS2 sea ice thickness will be underestimated if snow freeboard is assumed and full or partial penetration of the radar wave into the snow layer does occur. Price et al. (2019) found that the sea ice thicknesses in McMurdo Sound obtained by either assuming snow or ice freeboard
using Eq. (1) or (2), respectively, could produce a difference in thickness of 1.7 m. The sensitivity of the derived sea ice thickness to variable penetration depths into the snow layer was also assessed in that study.

Given considerable variability in the composition of the snow, particularly in the east where more wind-compaction was observed, it is likely that the penetration of the radar waveform into the snow layer will vary along-track. However, we
endeavoured to identify the best freeboard interface using the in situ measurements as opposed to assuming some arbitrary or constant freeboard interface which would have weakened the results. To ensure that the best-matching freeboard interface or penetration factor for each track was selected, we compared calculated CS2 ice thicknesses from equations 1, 2 and 3 with

interpolated drill hole MET along-track. Additional validation for the selected freeboard interface was provided by comparing mean values of CS2 freeboard, and interpolated drill hole sea ice and snow freeboard for each individual track. Importantly, our analyses compared linear trends and spatial patterns in CS2 ice thickness with respect to interpolated in situ measured fast ice and SPL thicknesses, and their combined MET as opposed to absolute values.

In the west and centre, the sea ice surface was the dominant freeboard interface, except in years with deeper snow coverage. In the east, the best matching freeboard horizon was variable from year to year and ranged from the snow surface to the sea ice surface or a mid-depth penetration. The mid-depth penetration into the snow in 2011 and 2018 agreed with the dominant ESA Level 2 backscattering horizon identified between the surface of the snow and the sea ice in Price et al. (2015) and is comparable to the upper limit penetration depth of 0.05-0.10 m estimated across the sound by Price et al. (2019) in November 2011.

This general pattern reflects the distribution and composition of the snow in McMurdo Sound. The snow in the east was generally deeper, more densely-packed, and wind-compacted relative to the centre and west where the snow was sparse, loosely-packed and where full penetration of the radar wave would more likely occur. The contrasting distributions in the snow and thicker ISW influenced fast ice and SPL from east to west was advantageous. We had confidence that the trends of higher freeboard and thicker CS2 obtained ice thickness observed in the centre relative to the east did not result from the addition of the snow layer to ice freeboard which would have a more significant effect in the east.

The freeboard to thickness conversion will also be affected by applying constant values for the density of sea ice, snow and seawater. However, error propagation analyses of the effects of the density values of sea ice, snow and seawater in the hydrostatic equilibrium equation carried out by Price et al. (2013); (2014) found that the assumed freeboard interface contributed the greatest error in the derived ice thickness.

## 5.3 CryoSat-2 satellite altimeter detection of ISW influenced fast ice

The objective of this study was to use a standard satellite elevation product (ESA Level 2 Baseline C SARIn product) and existing methods to obtain fast ice freeboard (Price et al., 2015) to assess, for the first time, whether a satellite altimeter is capable of consistently detecting a known pattern of higher freeboard driven by supercooled ISW outflow in McMurdo Sound.

Novel aspects of this study are:
1) The application of satellite altimetry to identify a known pattern of higher freeboard caused by thicker ice shelf-influenced fast ice combined with the buoyant forcing of the SPL which both result from supercooled ISW outflow.
2) Calculation of ice thickness from CS2 freeboard in McMurdo Sound for multiple years and comparison with interpolated in situ measured ice shelf-influenced fast ice and SPL and their combined MET.
3) Assessment and comparison of regional trends in CS2 freeboard (Table 1), and linear trends in CS2 ice thickness towards the McMurdo Ice Shelf in a region with significant ISW influence (centre) and another region with less pronounced ISW influence (east) (Fig. 4).
4) Assessment of spatial patterns of CS2 freeboard and CS2 ice thickness and comparison with in situ observed distributions of ice shelf-influenced fast ice and SPL, presented as a combined MET in Fig. 5.

CS2 conclusively detected the influence of ISW on fast ice in McMurdo Sound each study year. The spatial distribution of higher CS2 obtained freeboard and thicker CS2 ice concurred with the distributions of thicker ice shelf-influenced fast ice and

SPL and prior observations of ISW and effective negative ocean heat flux in McMurdo Sound (Barry and Dayton, 1988; Dempsey et al., 2010; Langhorne et al., 2015; Lewis and Perkin, 1985; Robinson et al., 2014). In the centre, CS2 freeboard and CS2 ice thickness increased concurrently with increasing fast ice and SPL thicknesses towards the ice shelf in the south on all the central profiles. The mean CS2 freeboard value in the centre over the main path of supercooled ISW outflow was 480 0.07 m and 0.09 m higher than the west and east, respectively, where the influence of ISW is less pronounced. The regional mean interpolated drill hole SPL thicknesses in the centre of McMurdo Sound was 3.90 m. The magnitude of this freeboard difference agrees with the 1-2 cm increase in freeboard for every metre of SPL obtained by Gough et al. (2012) using thermistor probe data. This higher freeboard centred at longitude ~165°E was also observed in McMurdo Sound by Price et al. (2013) with ICESat-1 over multi-year ice. The regional mean CS2 ice thickness in the centre was 0.78 m or 35 % greater than the 485 mean drill hole measured sea ice thickness. We mainly attributed this overestimate in satellite altimeter derived ice thickness to the additional buoyant forcing of the SPL.

**5.4 Outlook for satellite altimetry detection of ice shelf-influenced fast ice freeboard**

Here, we made the first steps towards developing a satellite-based method to identify and constrain regions of Antarctic coastal 490 sea ice that are being influenced by ISW outflow in the upper surface ocean. 'Potentially' supercooled ISW in the upper surface ocean promotes sea ice formation and causes it to be thicker by stabilising the upper surface layer, by hindering vertical mixing and insulating sea ice from warmer subsurface waters below (Hellmer, 2004). Thicker Ice Shelf Water influenced sea ice inherently has a higher freeboard than sea ice without this influence. Platelet ice is a direct manifestation and distinct signature of 'in situ' supercooled ISW at the ocean surface which causes the fast ice freeboard to stand higher through two effects: 1) 495 the fast ice is thicker by platelet ice consolidation, and augmented growth through heat flux into the ocean, and 2) the buoyancy effect of the SPL, if present. Satellite altimetry measurements cannot differentiate between higher freeboard driven by 'in situ' or 'potentially' supercooled ISW but as demonstrated here, could identify a region where the fast ice freeboard near an ice shelf has significantly higher than average freeboard due to the influence of ISW in the upper surface ocean.

Regions of anomalously higher freeboard could indicate thicker fast ice and the influence of ISW. This would require prior knowledge of the age and formation conditions of the fast ice (e.g., deformed ice will be thicker) and most critically the depth of the snow layer, if present. To improve uncertainty surrounding the radar altimeter penetration and snow depth CryoSat-2 could be used in tandem with ICESat-2 laser altimeter which measures snow freeboard. Once regions of ISW outflow are identified, long-term interannual variability in ice shelf and sea ice interactions could in theory be monitored with satellite 505 altimetry. The presence and abundance of both consolidated platelet ice and the SPL provide some insight into the processes at play within inaccessible ice shelf cavities, and the volumes of ISW outflowing in a region (Langhorne et al., 2015). With long-term monitoring, this could provide information on the effects of variability in atmospheric and oceanographic interactions which drive ISW formation within ice shelf cavities.

Here, we have demonstrated a method with the potential capability to carry out satellite altimetry assessments of regions where ISW and platelet ice have already been detected at the surface, and to identify other unknown regions where ISW is reaching the upper surface ocean. The smooth gradients in fast ice and SPL thickness and a low snow coverage in McMurdo Sound present favourable conditions for the CS2 radar altimeter to detect higher ice shelf-influenced freeboard. However, more challenging conditions for satellite altimetry are likely to be presented elsewhere on the Antarctic coastline. A recent drill hole 515 assessment of supercooled ISW-influenced fast ice in Atka Bay observed deep snow accumulations of up to 0.89 m which resulted in frequent negative fast ice freeboard, regardless of a substantial SPL beneath (Arndt et al., 2020). As far as we are aware, Atka Bay and McMurdo Sound are the only two locations on the Antarctic coastline with multiple years of in situ

measurements of ice shelf-influenced fast ice, SPL and snow highlighting the need for a satellite-based method to identify other regions where ISW is outflowing in the upper surface ocean and influencing fast ice.


However, significant challenges are presented, notably the noise in the CS2 measurement, interference from land, inaccuracies of geophysical corrections, the identification of relative SSH, the range resolution of SARIn, and most critically inadequate knowledge of the snow distribution in Antarctica and penetration depth of the radar waveform. Recently, a retracking algorithm was developed with the capability to retrieve both the backscattering horizon for the air to snow and snow to sea ice interface

from the CryoSat-2 Level 1b waveform (Fons and Kurtz, 2019) which showed significant promise. However, ice freeboard was overestimated in regions of large snow depths which concurred with a similar effect on CS2 obtained freeboard in deep snow deposits on Arctic sea ice (Ricker et al., 2015).

**6 Conclusion**

The outflow of supercooled Ice Shelf Water (ISW) from the conjoined McMurdo-Ross ice shelf cavity results in a consistent

pattern of thicker fast ice with a substantial sub-ice platelet layer (SPL) in the central-western region of McMurdo Sound. The thicker fast ice and the buoyant forcing of the SPL result in higher freeboards. Here, we investigated if the CryoSat-2 satellite radar altimeter is capable of detecting this higher freeboard. CryoSat-2 ice freeboard was obtained from surface elevation measurements by applying a supervised retrieval procedure which manually identified the relative sea surface height along-track in satellite imagery. CryoSat-2 ice freeboard was then compared to four years of drill hole measured ice and snow

freeboard, sea ice and SPL thicknesses, and snow layer depths on the fast ice in McMurdo Sound.

The spatial distribution of higher CryoSat-2 derived ice freeboard concurred with the distribution of thicker ice shelf-influenced fast ice and the SPL in late spring of 2011, 2013, 2017 and 2018. In the centre of McMurdo Sound, increasing trends in CryoSat-2 obtained freeboard and ice thickness were observed with increasing fast ice and SPL thicknesses towards the

McMurdo Ice Shelf every year. Over the four study years, we observe a mean CryoSat-2 obtained freeboard difference of 0.07-0.09 m across the main path of ISW outflow in McMurdo Sound. We interpret this freeboard distribution as being largely due to the thicker ice shelf-influenced fast ice and the substantial SPL across the main path of supercooled ISW outflow, rather than due to gradients in snow thickness. CryoSat-2 derived ice thickness were 35 % greater than drill hole measured fast ice thickness in the centre of the sound which we mainly attribute to the additional forcing of the SPL which had a regional mean

thickness of 3.90 m.

Several important factors complicate the identification of fast ice freeboard measured by satellite altimeters including inadequate knowledge of the snow layer in Antarctica, lack of adjacent open water nearby, the identification of a relative sea surface height and inaccuracies in the modelled geophysical corrections and geoid surface. We were able to constrain these

uncertainties and have confidence in the retrieved CryoSat-2 freeboard given the availability of in situ information for validation. The geophysical corrections applied to the CryoSat-2 Level 2 SARIn product and the effects of de-trending for the geoid over ice-free open water in McMurdo Sound were assessed during the study period to provide confidence in the CryoSat-2 measured freeboard and derived ice thickness.

Smooth gradients were observed in fast ice and SPL thicknesses and the snow layer providing favourable conditions for a satellite altimetry assessment. The thinner and generally non-compacted snow layer in the centre, west and northwest, where the influence of Ice Shelf Water is most pronounced, aided the detection of ice shelf-influenced freeboard with the CryoSat-2 radar altimeter, as it reduced the complication with uncertainty of penetration depth into the snow layer and interpretation of

the assumed backscattering/freeboard interface. It is possible that many unobserved regions of coastal sea ice around Antarctica

are influenced by the outflow of ISW in the upper surface ocean and the presence of platelet ice. We have shown that the CryoSat-2 satellite radar altimeter is capable of detecting higher freeboard driven by supercooled ISW outflow in McMurdo Sound and provide a proof-of-concept demonstration for the potential wider application of this method with adequate information on the snow layer.

**Appendix A**

The spatial distribution and magnitude of the geophysical corrections applied to the CS2 Level 2 SARIn product and the effect of de-trending for the geoid using the Earth Gravitational Model 2008 (EGM 2008) (Pavlis et al., 2012) were assessed over ice-free open ocean between latitudes 77.4° S and 78° S in McMurdo Sound in late summer (February to March) of 2011 to 2017. The date of minimum fast ice extent in late summer of each study year and days thereafter when McMurdo Sound was

free of sea ice were identified in MODerate resolution Imaging Spectroradiometer (MODIS) optical satellite images. The CS2 Level 2 SARIn derived sea surface with the *MSS*, and all ocean/tidal and atmospheric corrections applied was obtained without the presence of sea ice. Atmospheric corrections were always applied. The spatial distribution and magnitude of the ocean/tidal and *MSS* corrections were then constrained over open ocean. The following surfaces in the study region were compared:

1. CS2 height with *MSS* applied and all ocean/tidal corrections applied.
2. CS2 height with *MSS* not applied and all ocean/tidal corrections applied.
3. CS2 height with *MSS* applied and all ocean/tidal corrections not applied.
4. CS2 height with *MSS* not applied and all ocean/tidal corrections not applied.
5. Earth Gravitational Model 2008 (EGM 2008) surface.
6. CS2 height with *MSS* applied and all ocean/tidal corrections applied (surface 1) and then de-trended for EGM 2008.

Repeating spatial patterns in the Level 2 SARIn product ocean corrections, *MSS* and CS2 height were observed from year to year. The ocean/tidal corrections were small in magnitude (cms), and when not applied introduced more small-scale variability (cms) in the sea surface. Figure A1 shows an along-track comparison of surfaces 1, 2, 5 and 6 from a CS2 track on 17 March

2011 over open ocean and thin nilas sea ice in McMurdo Sound. The flattest profiles were obtained every year from the CS2 height with the *MSS* applied and all ocean/tidal corrections applied (surface 1). It was concluded that the geophysical corrections applied to the CS2 Level 2 SARin product were of good quality and resulted in reliable CS2 heights in McMurdo Sound.

Applying the MSS correction should account for the shape of the geoid surface. However, we observed that the geoid is not accounted for in the shape and magnitude of the CS2 height in the 7 years of CS2 measurements assessed over open ocean. We would expect surface elevation retrievals of an approximate magnitude of -2 m relative to WGS84 ellipsoid (Figure A1b) when the geoid surface has been removed and not the observed ~-55 m (Figure A1a). De-trending the CS2 height from surface 1 (with all corrections applied including the *MSS*) with the EGM 2008 geoid produced the flattest along-track profiles between

latitudes 77.4° S and 78° S in McMurdo Sound (surface 6). This pattern was consistent for the Baseline-C Level 2 SARIn product across McMurdo Sound every year, over open water in late summer, and over the fast ice in late spring (November). Additionally, freeboard profiles obtained from the CS2 Level 2 SARIn product for surface 6 agreed and aligned with both the magnitude and trends in drill hole measured ice and snow freeboard on fast ice in McMurdo Sound in late spring.

To summarise, the flattest profiles every year in McMurdo Sound were obtained from the CS2 height with the *MSS* applied and all ocean corrections applied, and then de-trended for the EGM 2008 geoid (surface 6). We were unable to clarify why the

CS2 Level 2 SARIn product with atmospheric, ocean corrections, and the *MSS* applied produced the flattest profiles when additionally de-trended for the EGM 2008 geoid. The *MSS* model applied in the study region is unclear from the information provided by ESA and is either the Aviso CLS01 (Webb and Hall, 2016; Bouzinac, 2012) or CLS 2011 (Skourup et al., 2017).

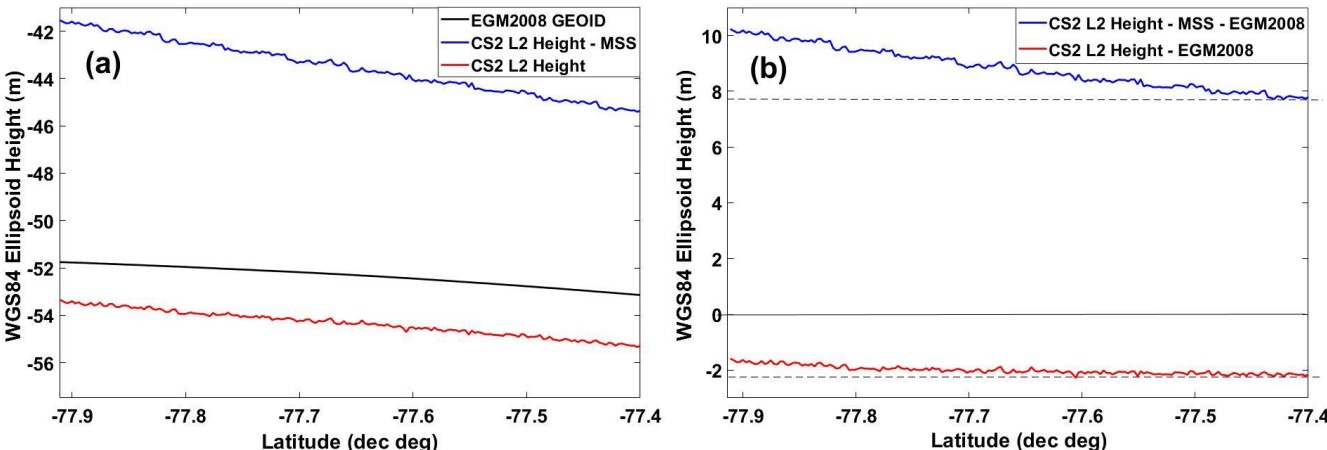

**Figure A1: Comparison of CryoSat-2 heights from the Level 2 SARIn product and the effects of de-trending with the Earth Gravitational Model 2008 (EGM 2008) geoid (Pavlis et al., 2012) over 55 km of open ocean in McMurdo Sound on 17 March 2011. (a) Surface 1: CS2 height with the *MSS* applied and all ocean corrections applied (red); Surface 2: CS2 height with the *MSS* not applied and all ocean corrections applied (blue); and Surface 5: EGM 2008 surface (black). (b) CS2 height with the *MSS* not applied and de-trended for the EGM 2008 geoid (blue), and Surface 6: CS2 height with the *MSS* applied and all ocean corrections applied and then de-trended for the EGM 2008 geoid (red). All heights are given relative to the WGS84 ellipsoid. Dashed horizontal lines are included in b to highlight the different gradients in the profiles.**

**Data availability.** The European Space Agency Level 2 Baseline C SARIn mode product (SIR_SIN_L2) was obtained at http://science-pds.cryosat.esa.int/ last accessed on 6 March 2019. The in situ data is available at the World Data Center PANGAEA for 2011 at https://doi.pangaea.de/10.1594/PANGAEA.933079, 2013 at https://doi.pangaea.de/10.1594/PANGAEA.933078, 2017 at https://doi.pangaea.de/10.1594/PANGAEA.933076 and for 2018 at https://doi.pangaea.de/10.1594/PANGAEA.933050.

**Author contributions.** GB collected in situ data in 2017 and 2018, designed the methodology, carried out all satellite data processing and analysis, and wrote the manuscript with input from all co-authors, in particular DP who significantly contributed to all stages of this study. DP wrote the base scripts modified by GB to process the CryoSat-2 Level 2 SARIn data. WR contributed significantly to the development of the methodology and manuscript editing, and WR and DP collected in situ data in 2011 and 2013. PL contributed to the manuscript development, coordinated the field programs and collected in situ data in 2011, 2013 and 2017.

**Competing interests.** The authors declare that they have no conflict of interest.

**Acknowledgments, Samples, and Data sources.** This research was funded by the Deep South National Science Challenge, the New Zealand Ministry of Business, Innovation and Employment through the Antarctic Science Platform (ANTA1801), the Marsden Fund Council from Government funding, managed by Royal Society Te Apārangi, a University of Canterbury Doctoral Scholarship, and logistics support provided by Antarctica New Zealand under K063 (2011, 2013, and 2018) and K066 (2016, 2017 and 2018) events. This research was carried out at Gateway Antarctica, University of Canterbury, in New Zealand. We express our deepest gratitude for the invaluable support provided by all K063 and K066 field event members and Scott Base staff, in particular to Dr. Greg Leonard from the University of Otago for his significant contribution to field work. We greatly appreciate the comments from the reviewers and editor Melody Sandells, which considerably improved the manuscript. We acknowledge the use of imagery from the NASA Worldview application

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
