# Peer review of "Satellite altimetry detection of ice shelf-influenced fast ice"

_The Cryosphere, 2020_

## Referee Comment (RC1) · Anonymous Referee #1 · 1 Dec 2020

General Comments

The article and language are clear and easy to follow and it is an interesting demonstration of the use of CryoSat-2 to retrieve information about ice shelf-influenced fast ice using satellite remote sensing. It would be helpful to briefly set out the significance of the detection of ISW in terms of the remote sensing and climate impacts this study and similar studies could have, by outlining which parts of the method are novel and the potential insights to be gained. (Insights are discussed in Section 5.4 but would be useful summarised briefly in the introductory remarks/study motivations.)

It is positive that the study considers the potential impact of the dominant backscattering surface being somewhere other than the upper ice surface, this could be expanded to include further quantification relating to snow conditions. There is a heavy reliance

on the assumption of hydrostatic equilibrium holding which is true for entire floes. Using this assumption requires careful sampling and the way this has been conducted in this study needs to be explained (point-to-point measurements an infrequent sampling over long length scales may not characterise the floe sufficiently well enough for an assumption of hydrostatic equilibrium to hold using measured thicknesses and densities.) Other studies looking at this area have found that sediments are present in large amounts which could also affect the ice and should be commented on (Rack et al., 2013 and Glasser et al, 2017) – it is important to outline whether this could this affect the assumptions used in the study. A constant ice density from the literature is assumed in this study – it would be good to determine the sensitivity of the conclusions to this considering its uncertainty, variability and validity for this study.

The methodology is sometimes not sufficiently detailed to assess what has been done, for instance there are many mentions of spline fitting without discussing the order or justification for the choice, noted in the Specific Comments. It is not clear in the text which aspects of the methodology are novel. These make it difficult to assess in terms of contribution to the field and quality of methodology.

Specific Comments

Line 57-58: It would be helpful to briefly outline the physical characteristics behind the buoyant forcing and how these influence ice freeboard.

Line 85: Please state which aspects of this methodology are novel.

Line 124: Are there other potential influences on the freeboard estimates, either from CS2 or in situ, ie not the SPL buoyancy effect. It would be good to confirm and justify as this is so crucial for the study. It would be good (either here, or further down) to justify the sampling strategy which ensured that hydrostatic equilibrium could be assumed, given that the whole floe will be in equilibrium whilst point measurements at limited locations may not indicate this. The spline fits cover huge areas but appear to be based on limited measurements.

Line 144: Please explain the spline interpolation including the order and a justification for this choice. Similar to the previous comment, are the drill holes single point measurements at each location and what is their uncertainty – and how does this compare to the lateral variation across a floe, and the justification for using hydrostatic balance to relate these quantities?

Line 160: Please explain the choice of a 'lower to mid-range SPL solid fraction'

Line 172: I have seen the SAR Interferometric mode sometimes referred to as SARIn, and sometimes (including here) as SIN. I don't know if the editor would have a preference for use in this journal?

Line 187: Please briefly justify whether you think the snow and ice characteristics are similar to November 2011, to explain the likely relevance or differences in comparison with your data.

Line 198-199: Please state why this is only in general (do you mean for your study or for others?) and how close the open water is relative to the study location – the distance is likely to be important in being able to compare local sea level.

Line 215: Please explain how well you think these corrections will have improved the freeboard estimates, can you give the level of remaining uncertainty and its causes? (Especially given the information in lines 523-532 and uncertainty about geiod detrending.)

Figure 2: It appears there may be a periodicity in the CS2 freeboards, is there a reason this might be the case and could it relate to corrections not entirely removing other effects in these data?

Line 255-257: It is a useful and interesting insight to see that the 'freeboard interfaces' were variable. Can you relate this to snow properties such as density if any data are available (you mention snow depth – can you quantify the effect of snow depth – is there a threshold value above which the ice surface is no longer the dominant scattering

surface or do you think other variables are influencing this also?) You mention this in Section 5.2 but do not quantify, but as you mention in Section 5 knowledge of the snow characteristics will be crucial.

Line 380: Sea surface height can vary over 25 km scales – it will be important to justify why this is not a problem for this study (the following sentences discuss this but it is not clear how this can be discounted as a source of bias.)

Line 427-431: Please quantify these so that a comparison can be made by specifying the increase you observed, using the SPL thickness measurements, to demonstrate how close the agreement is. Please include the magnitude of the higher freeboard in Price et al. (2013) and give an indication of the magnitude of the additional buoyant forcing of the sub-ice platelet layer.

Technical Corrections

Line 167: It would help to show individual tracks if they were plotted with slightly slimmer lines, especially for the East.

Figure 3: Missing label for x-axis

Line 355: Should 'd' be followed by a bracket ')'?

---

## Referee Comment (RC2) · Anonymous Referee #2 · 3 Dec 2020

Fast ice is an important component of the Antarctic climate system, especially for a better understanding of processes between ice shelf and ocean. It is therefore out of question that there is a great interest in the community to obtain better estimates of the thickness of the solid ice over the whole area and thus to be able to conduct further calculations of sea ice volume in the next step. However, the manuscript still shows fundamental weaknesses at this stage, so that it needs to be extensively revised before the work can be published.

That is why I only mention here general remarks, which should be implemented first, before in a next iteration more detailed points can be raised.

General Comments:

[Figure]

1. I see a major mismatch between the title, the objectives and the actual content of the manuscript. What is basically done in the manuscript is to compare measured on-site freeboard values related to fast-ice and platelet ice thickness with satellite retrieved freeboard values. However, this does not reflect the actual influence of ISW on the fast ice, as ISW is not necessarily the same as platelet ice.

2. Therefore, a fundamental revision of clearly stated objectives is required, which are then also addressed accordingly in the manuscript.

3. All applied methods in the manuscript are very poorly presented, so that it is not always 100% clear what was really done in detail. Therefore, in order to be able to judge the exact quality and reliability of the analyses presented, they must be made clearer. Furthermore, there is no clear distinction between the work/analyses presented here and what was done in previous work.

4. The snow cover on Antarctic sea ice is known to play a crucial role in both remote sensing and the buoyancy principle of sea ice. Even though the time series shown here seems to have a negligible snow thickness, this cannot be neglected in such a study. Instead, much more attention must be paid to potential difficulties caused by superimposed ice, snow ice or severe snow metamorphism.

5. Moreover, this very thin layer of snow raises serious doubts as to the extent to which such a study can be used beyond the study area shown here. Indeed, a stronger positive freeboard of fast ice areas is not only due to the buoyant platelet layer. Instead, studies, e.g. in Atka Bay (Arndt et al., 2020), have shown that it is not the platelet ice that is the decisive component for the freeboard of the fast ice, but the snow cover. I therefore strongly recommend to do similar sensitivity studies for this region - and also to use data from previous years to emphasize that in McMurdo Sound there is always this low snow load. Even if this is the case, the conclusions must still be strongly weakened, because a strong positive freeboard can have different reasons - which cannot be quantified with CryoSat alone.

6. Referring to the previous point, the work shown here would greatly benefit from putting the measured snow, fast ice and platelet ice thicknesses into a more global context with measurements from other regions or other points in time in the same region. Otherwise, the results found here are unfortunately not very reliable and raise great doubts that they can be applied on a larger scale.

---

## Referee Comment (RC3) · Anonymous Referee #3 · 7 Dec 2020

General Comments. The paper examines the ability of CryoSAT 2 to detect the existence of a sub-ice platelet layer found under McMurdo Sound fast ice and previously determined to result from the upwelling of supercooled Ice Shelf Water. The study provided good comparisons with the freeboard rise from satellite altimetry with a detailed ground truth campaign conducted over four years of measured sea ice freeboard, snow depth, sea ice thickness and sub-ice platelet layer thickness distributions. Proof of the utility of satellite altimetry to effectively determine the distribution of these sub-ice platelet layers even in a selected region like McMurdo Sound is worthy of publication given that the lack of ground truth in other sea ice studies is a continuous impediment to progress in maximizing the potential of satellite remote sensing to effective monitoring of sea ice processes from space. There are some needed revisions to fully realize

the paper's potential.

Specific Comments. Proof of the utility of satellite altimetry to determine the distribution of these sub-ice platelet layers will provide an effective means of monitoring them from space and help in monitoring the interannual variability of the flux of Ice Shelf Water from underneath the Ross and McMurdo Ice Shelves in future. That said, there are some difficulties in the presentation of the results. For example, in Figure 5, the use of the same color bars for quite different scales, CS2 freeboard in 5a (up to 0.6m), CS2 ice thickness in 5b (up to 5.8m) and drill hole MET in fig 5c (up to 3.5m) is difficult to interpret correctly. (note also in Technical Comments about the need for intermediate values). In the abstract the sentence "We demonstrate the capability of CryoSat-2 to detect higher Ice Shelf Water influenced fast ice freeboard in McMurdo Sound and the wider application of this method as a potential tool to identify regions of ice shelf-influenced fast ice elsewhere on the Antarctic coastline." Is a reach too far, given the unique condition of McMurdo with its generally very thin snow cover which may not be generally found in other coastal regions. There is also no attempt in the paper itself to apply the technique to other regions. Suggest limiting the statement to only: "We demonstrate the capability of CryoSat-2 to detect higher Ice Shelf Water influenced fast ice freeboard in McMurdo Sound." In the discussion of the paper, the concepts can be best given as to applicability to other regions, with sufficient caveats given as to the role of thicker snow than found in McMurdo Sound for example, and how this may affect the interpretations elsewhere.

Technical Comments. Abstract: "We attribute this overestimate in satellite altimeter obtained ice thickness to the additional buoyant forcing of the sub-ice platelet layer. " Comment: Need to know if the measurement of the sub-ice platelet layer distribution verified this. Line 124: What were the equivalent freeboard rises in cm to the 12% and 19% freeboard increases at those locations? Were there also thickness measurements of the SPL at that time? Line 139 (Grammar error)."Refer to Brett et al. (2020) for a detailed description of the thickness distributions of ice shelf-influenced fast ice, the

SPL and snow in McMurdo Sound in November of 2011, 2013 and 2017." Change to: Brett et al. (2020) provide a detailed description of the thickness distributions of ice shelf-influenced fast ice, the SPL and snow in McMurdo Sound in November of 2011, 2013 and 2017.

Line 140 Change to: Here we summarize those descriptions to show general patterns and also include the fast ice conditions in 2018. Line 146 give a value for more substantial deposition of snow Figure 1. Give some intermediate SPL values on the color bar rather than just the High and Low. Might also include a few (4 or 5) of these as identified contour lines on the map plot. In the caption point out that the red square on the inset map is the area (McMurdo Sound) shown on the MODIS image to the right. Figure 5 need intermediate values on the color bar (Freeboard, C2-2 Ice thickness, Drill Hole MET) and address concerns about the same color bars but different scales for a, b and c. Line 490 Appendix A Seems hard to read, Line 530 This indicates that the geoid is de-trended for twice (?? Don't understand this sentence)

Lines 516-532 (Appendix A). I find this discussion rather confusing, perhaps hampered by my own limited knowledge of Geodesy. For example, the sentence (Line 507) "The open ocean surface with MSS and ocean/tidal corrections removed was consistent from 2011 to 2017." Is this both MSS and ocean/tidal corrections removed (Item 4) or Is it Item 3, MSS applied and ocean /tidal corrections removed? I infer that the authors' find the best practice for surface elevation is Item 6. in their first list but this is difficult to suss out from their discussion. Suggest a Table listing the various options and some index of performance e.g. Good, Fair , Poor along with better referencing of the number of the option in the accompanying text would help to clarify.

Line 542 "The in situ data included in this study will not be available at the time of publication but it is intended that it will be deposited in a data repository." However from the Journal Data Policy:"Copernicus Publications requests depositing data that correspond to journal articles in reliable (public) data repositories, assigning digital object identifiers, and properly citing data sets as individual contributions. . ... Authors are required to provide a statement on how their underlying research data can be accessed. This must be placed as the section "Data availability" at the end of the manuscript." Reviewer Comment: Further revisions should include the citation to Data Availability required by this journal. A substantive further review may require the reviewers' and editor to examine the data used in the paper before final approval and the paper cannot be examined by others without the ability to further examine the data and conduct their own analyses.

---

## Author Comment (AC1) · 26 Jan 2021

Authors Response to Anonymous Referee #1

Authors: We thank the referee for taking the time to review our manuscript and for providing valuable feedback. We have considered your comments and modified the manuscript according to suggested changes where we agree and provided a justification where we do not. We hope that the responses given below and modifications made have addressed the reviewer's comments.

Referee 1: General Comments The article and language are clear and easy to follow and it is an interesting demonstration of the use of CryoSat-2 to retrieve information about ice shelf-influenced fast ice using satellite remote sensing. It would be helpful to

briefly set out the significance of the detection of ISW in terms of the remote sensing and climate impacts this study and similar studies could have, by outlining which parts of the method are novel and the potential insights to be gained. (Insights are discussed in Section 5.4 but would be useful summarised briefly in the introductory remarks/study motivations.)

Author Comment: We thank the reviewer for their suggestion and have made the following changes to the introduction:

1) To highlight the potential implications and significance of ISW detection with satellite remote sensing, we have we have restructured the introduction and combined it with the first sub-section of section 2.

We made the following changes: - moved the sentence on L96 to L70; - moved paragraph L102-114 to L73 and added the following sentence after Line 114:

'Using ground measurements, Brett et al. (2020) demonstrated a correlation of SPL thickness and volume in late spring with a higher frequency of strong southerly wind events in the western Ross Sea which drive polynya activity, HSSW production and ISW formation and circulation within the McMurdo-Ross ice shelf cavity over winter.'

- And additionally, we have added the following statement to L76:

'A means to identify these regions in large-scale satellite assessments is highly desirable, and if effective, has the potential to provide a satellite-based method to monitor the interactive system at play between the atmosphere, coastal polynyas, and circulation within ice shelf cavities with further development. The detection of ISW influence on fast ice via satellite altimetry is in theory possible through the identification of regions with 'anomalously' higher freeboard driven by combination of thicker ice shelf-influenced sea ice and the buoyant forcing of a SPL, if present (Price et al., 2014) but to date has not been assessed.'

- The two sentences on L97-100 have been moved to L127

2) To emphasise that the main method used in the study has been developed in detail in previous satellite altimetry work in McMurdo Sound, we have moved the paragraph in section 2 (L115-124) to L85 and added the following sentence at the end of this paragraph:

'Price et al. (2015) developed the method applied in this study to obtain CS2 fast ice freeboard in McMurdo Sound in 2011 and 2013 and the relevance of this work to this study is described in more detail in section 2.2'

Referee 1: It is positive that the study considers the potential impact of the dominant backscattering surface being somewhere other than the upper ice surface, this could be expanded to include further quantification relating to snow conditions.

Author Comment: Thank you for this comment. However, to quantify and constrain the effects of the snow layer on the CryoSat-2 radar waveform, one must carry out highly detailed assessments of the dielectric properties of the snow cover including grainsize, layering, brine content, temperature profiles and snow metamorphosis. This detailed snow information was not collected in our field campaigns. Price et al. 2015 have already assessed the effects of the geometric and radar roughness of the snow layer, snow depth, density and grain size on the returning radar waveform and freeboard retrievals from three different retrackers in McMurdo Sound in late spring of November 2011 and 2013.

As described in L183-185 (of this study), Price et al. 2015 found that the ESA Level 2 retracker tracked between the snow surface and snow-ice interface. They also state that one would need to know the backscattering coefficient of snow and ice to quantify what impact the snow cover has on the retrieved freeboard. In this study, the best we can do is relate the backscattering interface to interannual variability in snow depth and wind-compaction which we do on L256-257; L301-303; 306-307; L406-417 but this can only be qualitative for reasons given in the response to the specific comment about L255-257 below.

Referee 1: There is a heavy reliance on the assumption of hydrostatic equilibrium holding which is true for entire floes. Using this assumption requires careful sampling and the way this has been conducted in this study needs to be explained (point-to-point measurements an infrequent sampling over long length scales may not characterise the floe sufficiently well enough for an assumption of hydrostatic equilibrium to hold using measured thicknesses and densities.)

Author Comment: The sampling strategy at field sites has been described in detail in previous studies (e.g., Gough et al. 2012, Price et al. 2014) and we were initially reluctant to repeat that information in this study. However, we agree that the spatial distribution of freeboard measurements at field sites is important when assuming hydrostatic equilibrium and have now included the following statement after line 131 in section 2.1.

'At each site, five drill holes were made in the sea ice at the centre and end points of two 30 m cross-profile lines. Sea ice freeboard, snow depth and the thicknesses of sea ice and the SPL were measured at each drill hole using the technique described in Price et al., 2014 and then averaged to give a representative value over the 30 m by 30 m area.'

To the best of our knowledge, the fast ice assessed in this study is likely to be in or at least very close to hydrostatic equilibrium as it was not measured in close proximity to narrow pressure ridges or the coast. Importantly, we focussed on regional (kms) trends (Table 1 and Figure 3 and 4) and spatial patterns (Figure 5) in our comparison of in situ and CS2 freeboards and not on variability in the small-scale (10's or 100's metres). In multiple field seasons in McMurdo Sound, we observed smooth gradients in the sea ice, SPL and snow which are comparable on the scale of the CS2 footprint. Brett et al. 2020 assessed the drill hole measurements presented in this study for 2011, 2013, and 2017 with a ground-based electromagnetic induction (EM) device to measure coincident sea ice and SPL at a 10 m sampling interval. The high resolution EM measurements showed that the distribution of sea ice and SPL thicknesses (which

mostly determine the resultant freeboard) in McMurdo Sound have smooth gradients on the kilometre scale.

Referee 1: Other studies looking at this area have found that sediments are present in large amounts which could also affect the ice and should be commented on (Rack et al., 2013 and Glasser et al, 2017) – it is important to outline whether this could this affect the assumptions used in the study.

Author Comment: The Glasser et al. 2017 study focuses on the significant sediment load on the land-ice of the McMurdo Ice Shelf (as do Glasser et al. 2006) and not on the adjacent sea ice which is the subject of this study. We find no mention of land-fast sea ice in the Glasser et al. 2017 study.

Rack et al. 2013 predominately assessed the debris contribution to the density of the McMurdo Ice Shelf. They did calculate a theoretical debris load to account for the overestimate in fast ice thickness in the centre of McMurdo Sound. However, they also state that this could be explained by the contribution of sub-ice platelet layer buoyancy to higher freeboard. The sub-ice platelet layer contribution to overestimates in calculated sea ice thickness from anomalously higher freeboard in McMurdo Sound was validated and quantified in a later study by Price et al. 2014 which is described and referenced in this study.

It is unlikely that the windblown sediment load is significant for the land-fast sea ice in McMurdo Sound as it is typically first-year ice of 6-8 months age and would not have time to accumulate the comparable sediment loads observed on the adjacent McMurdo Ice Shelf (likely 1000's of years of accumulation of surface deposition and marine/anchor ice from the bottom). We only consider First-Year ice in the linear and spatial trend analyses in this study.

Importantly, sea ice grows downwards with congelation growth at the base of the ice, and additionally in McMurdo Sound with platelet ice consolidation at the sea ice base, and augmented growth through heat flux to the heat-deficit in the ocean driven by

supercooled Ice Shelf Water. This is in contrast to the typical mass contribution to land ice/ice shelves of surface snow accumulation where sediment can become more readily incorporated. We are aware that McMurdo Ice Shelf is 'unusual' in that basal freezing is a major contributor to its mass balance and that sediments entrained in the anchor/marine ice reach the surface of the ice shelf as it ablates.

Referee 1: A constant ice density from the literature is assumed in this study – it would be good to determine the sensitivity of the conclusions to this considering its uncertainty, variability and validity for this study.

Author Comment: Price et al. 2013 and Price et al. 2014 assessed error contribution from sea ice density in detail and provide a sound justification (based on a range of sea ice density measurements in McMurdo Sound) for the selected value of 925 kg m-3. To clarify that this has already been assessed in detail we have added the following sentence at L241:

'. . .to adhere to the rationale and error propagation in Price et al. 2014 where a range of snow and sea ice density measurements made in McMurdo Sound were assessed.'

Additionally, on L396-399 we referenced the findings of Price et al., 2013 and Price et al., 2014 who found in their detailed error analyses that the main error contribution to calculated ice thickness is the freeboard.

Referee 1: The methodology is sometimes not sufficiently detailed to assess what has been done, for instance there are many mentions of spline fitting without discussing the order or justification for the choice, noted in the Specific Comments.

Author Comment: In multiple field seasons in McMurdo Sound, we observed smooth gradients in the thickness of sea ice, SPL and the snow layer which were comparable on the scale of the CS2 footprint. We thus applied minimum curvature (i.e., thin plate) first derivative spline interpolations which pass through the data points and no smoothing. We will add this detailed description of the spline interpolation and the justification

for applying it to the discussion of the interpolation on L134-138.

Referee 1: It is not clear in the text which aspects of the methodology are novel. These make it difficult to assess in terms of contribution to the field and quality of methodology.

Author Comment: The motivation is to use a standard satellite elevation product (ESA L2 Baseline C SIN product) and existing and proven methods to obtain fast ice freeboard (as developed and demonstrated in Price et al. 2015) to assess whether a satellite altimeter is capable of detecting a known pattern of higher freeboard driven by supercooled ISW outflow in McMurdo Sound. To summarise, much of the method is not novel but the application is. We have made the changes described in response to the first general comment above to emphasise the previous work by Price et al. 2014 and 2015 that underpins the methodology applied in this study.

We will highlight that this is the first study applying satellite altimetry to specifically detect ice shelf-influenced fast ice freeboard by changing the statement on L85 to the following:

'For the first time, we investigate whether the CS2 satellite radar altimeter can detect the influence of ISW on fast ice in McMurdo Sound by consistently identifying the higher ice freeboard caused by thicker ice shelf-influenced fast ice combined with the buoyant forcing of the SPL beneath.'

We will also highlight and emphasise the methods that are novel throughout the text. Novel aspects of the study are as follows: - The application of satellite altimetry to identify a known pattern of higher freeboard caused by thicker ice shelf-influenced fast ice combined with the buoyant forcing of the SPL which both result from supercooled ISW outflow. - The method to identify the best-matching freeboard interface for individual CS2 tracks as described in L251-260 and L401-405. - Calculation of ice thickness from CS2 freeboard in McMurdo Sound for multiple years and comparison with interpolated in situ measured ice shelf-influenced fast ice and SPL and their combined Mass Equivalent Thickness (MET). L236-249; L290-327; Figures 4 and 5. - Assessment and

comparison of regional trends in CS2 freeboard (Table 1), and linear trends in CS2 ice thickness towards the McMurdo Ice Shelf in a region with significant ISW influence (centre) and another fast ice region with less pronounced ISW influence (east) (Figure 4). - Assessment of spatial patterns of CS2 freeboard and CS2 ice thickness and comparison with in situ observed distributions of ice shelf-influenced fast ice and SPL (presented as a combined Mass Equivalent Thickness (MET)) (Figure 5).

Referee 1: Specific Comments Line 57-58: It would be helpful to briefly outline the physical characteristics behind the buoyant forcing and how these influence ice freeboard.

Author Comment: This is stated in L59-63 and L150-161 as the thickness of the layer and the solid ice fraction which is then discussed in detail in the following paragraph. Price et al. 2014 assessed the contribution of the SPL and the solid ice fraction to freeboard in McMurdo Sound and this work is described and referenced several times in the text.

Referee 1: Line 85: Please state which aspects of this methodology are novel.

Author Comment: As above in response to the general comment. The first sentence of this paragraph L85-87 has been changed to the following:

'For the first time, we investigate whether the CS2 satellite radar altimeter can detect the influence of ISW on fast ice in McMurdo Sound by consistently identifying the higher ice freeboard caused by thicker ice shelf-influenced fast ice combined with the buoyant forcing of the SPL beneath.'

Referee 1: Line 124: Are there other potential influences on the freeboard estimates, either from CS2 or in situ, ie not the SPL buoyancy effect. It would be good to confirm and justify as this is so crucial for the study.

Author Comment: The mean 12% and up to 19% overestimation in sea ice thickness calculated by Price et al. 2014 were predominately driven by the sub-ice platelet layer

because in that study they specifically isolated the contribution of the SPL (and solid ice fraction) to the higher freeboard.

Ice Shelf Water influenced thicker fast ice also has an inherently higher freeboard. In this study, we identify higher freeboard driven by both the thicker ice shelf-influenced fast ice combined with the SPL buoyancy. We state this multiple times in the text including the abstract and in particular on L150-152. The CS2 measured freeboard is influenced by the snow which is relatively thin in McMurdo Sound and which we constrained to the best of our abilities using the in situ measurements as described. There is no evidence of other influences on the sea ice freeboard displayed in the region nor expected from the physical forces at play. Snow depressed negative freeboard or surface flooding was not observed, and we have added the following sentence on line 152 to clarify this:

'The snow layer can depress the freeboard and result in flooding of the sea ice surface and the formation of meteoric ice which can contribute to freeboard (Maksym and Markus, 2008). Snow-depressed negative freeboard or surface flooding were not observed at drill hole sites in McMurdo Sound in late spring. Multiple ice core studies carried out in the region over winter and in late spring revealed no contribution of meteoric ice to the fast ice cover in McMurdo Sound (e.g. Dempsey et al., 2010, Gough et al., 2012)'

We discuss how the snow layer in McMurdo Sound would affect the results of this study on L412-417 highlighting that the snow distribution from west to east was advantageous in that we could have confidence in that the trends in higher freeboard in the centre of the sound (where the snow is thin and loosely-packed) did not result from the addition of the snow layer which would have a more significant effect in the east where the snow is deeper.

We have also now included a discussion of the findings of Arndt et al., 2020 in section 5.4 on L435:

'The smooth gradients in fast ice and SPL thickness and low snow coverage in Mc-Murdo Sound present favourable conditions for the CS2 radar altimeter to detect higher ice shelf-influenced freeboard. However, more challenging conditions for satellite altimetry are likely to be presented elsewhere on the Antarctic coastline. A recent drill hole assessment of supercooled ISW-influenced fast ice in Atka Bay observed deep snow accumulations of up to 0.89 m which resulted in frequent negative fast ice freeboard regardless of the buoyant forcing of a substantial SPL beneath (Arndt et al., 2020). As far as we are aware, Atka Bay and McMurdo Sound are the only two locations on the Antarctic coastline with multiple years of in situ measurements of ice shelf-influenced fast ice, SPL and snow highlighting the need for a satellite-based method to identify other regions where ISW is present in the upper surface ocean and influencing fast ice formation.'

Referee 1: It would be good (either here, or further down) to justify the sampling strategy which ensured that hydrostatic equilibrium could be assumed, given that the whole floe will be in equilibrium whilst point measurements at limited locations may not indicate this. The spline fits cover huge areas but appear to be based on limited measurements.

Author Comment: Please refer to the responses given to general comments on the hydrostatic equilibrium assumption and justification for a spline interpolation.

Referee 1: Line 144: Please explain the spline interpolation including the order and a justification for this choice. Similar to the previous comment, are the drill holes single point measurements at each location and what is their uncertainty – and how does this compare to the lateral variation across a floe, and the justification for using hydrostatic balance to relate these quantities?

Author Comment: Please refer to the responses given to general comments on the hydrostatic equilibrium assumption and justification for a spline interpolation.

Referee 1: Line 160: Please explain the choice of a 'lower to mid-range SPL solid

fraction'

Author Comment: We chose this as an intermediary value between that of Gough et al. 2012 (0.25) and Price et al. 2014 (0.16) and have changed the sentence on L160 to the following to clarify this:

'To calculate MET from the interpolated drill hole measurements, an intermediary value of 0.2 between the values determined by Gough et al. 2012 (0.25) and Price et al. 2014 (0.16) was assumed for the solid fraction each year.'

Referee 1: Line 172: I have seen the SAR Interferometric mode sometimes referred to as SARIn, and sometimes (including here) as SIN. I don't know if the editor would have a preference for use in this journal?

Author Comment: We do not have a strong preference.

Referee 1: Line 187: Please briefly justify whether you think the snow and ice characteristics are similar to November 2011, to explain the likely relevance or differences in comparison with your data.

Author Comment: They are the same drill hole measurements and fast ice conditions assessed in November 2011 by Price et al 2019 and we have now clarified this on L187.

'Price et al. (2019) assessed the sensitivity of CS2 Level 2 SIN product derived sea ice thicknesses to variable penetration depths into the snow layer in McMurdo Sound using the same in situ measurements in late spring November 2011.'

Referee 1: Line 198-199: Please state why this is only in general (do you mean for your study or for others?) and how close the open water is relative to the study location – the distance is likely to be important in being able to compare local sea level.

Author Comment: We use the term 'In general' referring to this as a standard method used in satellite altimetry. To clarify, we have changed the sentence to the following:

'In satellite altimetry, freeboard is generally determined relative to a local reference sea surface height (SSH) obtained over open water along the satellite track.'

The distance on the fast ice to open water never exceeded 25 km and this is stated on L379-380. For clarity, we will state the following on L219:

'The distance on the fast ice to open water did not exceed ∼25 km in all study years.'

Referee 1: Line 215: Please explain how well you think these corrections will have improved the freeboard estimates, can you give the level of remaining uncertainty and its causes? (Especially given the information in lines 523-532 and uncertainty about geiod detrending.)

Author Comment: Detailed descriptions of the geophysical corrections are given in Bouzinac (2012) and Webb and Hall (2016) as stated on L200-202. In our study region, the magnitude of most of the geophysical corrections ranges from sub-centimetre to 1-2 centimetres and generally does not vary much over the small fast ice area. The Dynamic Atmospheric Correction is on the order of 10's cm but again varies very little in the small study area (<1 cm). The Mean Sea Surface varies from 8-11 m and is important to apply as it introduces a steep slope in the surface as demonstrated in Figure A1. We used the in-situ measurements to remove tracks that had erroneous profiles. This tended to occur close to high topography such as Ross Island and was rather significant and easy to identify. The in situ measurements were up to 10 km apart but the gradients in sea ice thickness and SPL are quite smooth in McMurdo Sound. The remaining uncertainty is likely driven by variability in the snow, the dominant backscattering horizon and noise in the CS2 measurement which should be captured in the standard deviations given in Table 1. To clarify, we add the following sentence to L377:

'The remaining uncertainty in the CS2 measurement, captured in the standard deviations given in Table 1, is likely to be driven by noise and variability in the snow layer, and the penetration depth of the radar wave along-track.'

Referee 1: Figure 2: It appears there may be a periodicity in the CS2 freeboards, is there a reason this might be the case and could it relate to corrections not entirely removing other effects in these data?

Author Comment: The assessment of the geophysical corrections in Appendix A was undertaken to ensure that the corrections were of good quality and produced along-track CS2 surface elevation profiles with minimal residual curvature remaining from geophysical effects. We are unsure why there is an apparent periodicity on this individual track. We examined the geophysical corrections along this specific track and found that the majority of corrections were of small magnitude (<1cm) and either a constant value was applied along-track or it varied very little (∼1 cm). This periodicity was not observed consistently in other tracks. The noise in the CS2 measurement is considerable and is impossible to account for. There was more snow and wind-compaction in this region in 2017 (L301-303) which may have contributed to the apparent signal in the CS2 freeboard in the track shown in Figure 2.

Referee 1: Line 255-257: It is a useful and interesting insight to see that the 'freeboard interfaces' were variable. Can you relate this to snow properties such as density if any data are available (you mention snow depth – can you quantify the effect of snow depth – is there a threshold value above which the ice surface is no longer the dominant scattering surface or do you think other variables are influencing this also?) You mention this in Section 5.2 but do not quantify, but as you mention in Section 5 knowledge of the snow characteristics will be crucial.

Author Comment: We thoroughly agree that this would be a very useful endeavour but as previously described, highly detailed information about the properties of the snow would be required to do this with any confidence. Even relating the backscatter behaviour to snow depth is ambiguous because snow layering or wind-compacted layers could play a more important role and this information was not collected in a systematic way except for in November 2011 and 2013. This has already been assessed and described in detail in Price et al. 2015. We are of the opinion that a detailed assessment of the snow is of enough importance to merit a separate study in itself, if the field information was available.

Referee 1: Line 380: Sea surface height can vary over 25 km scales – it will be important to justify why this is not a problem for this study (the following sentences discuss this but it is not clear how this can be discounted as a source of bias.)

Author Comment: We agree that it cannot be discounted as a source of bias but to ensure minimum error in the relative SSH, we carried out the following steps:

1. We first assessed the geophysical corrections (ocean, tidal, Mean Sea Surface and geoid) applied to the L2 SIN product in our study area (77.4-78° S) in Appendix A. We did this to ensure that the corrections were of good quality and produced along-track CS2 surface elevation profiles with minimal residual curvature remaining from these geophysical effects. This is stated on L371-376.

2. We then applied a supervised retrieval procedure where open water was manually identified in satellite imagery. Sea surface conditions such as waves will have a more significant effect for automatic retrievals which interpret the backscatter or pulse peakiness of the signal to identify open water as assessed and discussed in Price et al. 2015.

3. We then assessed the resultant CS2 freeboards through comparison with in situ measured freeboard. This is stated on L382-383.

We used 25 km of open water to calculate the median relative sea surface height because this distance provides ∼80 CS2 measurements from which to obtain a representative median value from the noisy CS2 measurement. The same 25 km distance was applied by Price et al., 2015 for their supervised freeboard retrieval which they assessed in detail. We will add the following sentence at L219:

'This distance provided approximately 80 CS2 surface elevation measurements to obtain a representative median value from the noisy CS2 measurement as demonstrated

by Price et al, 2015 in their supervised freeboard retrieval.'

Additionally, we have focussed on linear and spatial trends in CS2 freeboard and ice thickness in this study, e.g., Figures 4 and 5. Any small error in the relative sea surface height should not overly affect the trends because a constant median SSH value (obtained from the ~80 CS2 measurements) is subtracted from the CS2 surface elevation measurements over fast ice to obtain CS2 freeboard for each individual track.

To clarify, we have now rewritten the paragraph on L378-385 to the following

'The geophysical corrections should not have a major impact on obtained freeboard if the retrieval of the relative SSH is robust (Ricker et al., 2016). Identifying the relative SSH along-track is complicated by interference and noise introduced by sea surface conditions and by the presence of pack ice. We observed thin nilas or some pack ice beneath several CS2 tracks. However, sea surface conditions will have a more significant effect for automatic freeboard retrievals which interpret the backscatter or pulse peakiness of the returning radar signal to identify open water (Price et al. 2015). To ensure minimum error in the relative SSH, we applied a supervised freeboard retrieval procedure where open water was manually identified in satellite imagery. The distance on the fast ice to open water did not exceed ~25 km in all study years. We used in situ measurements to assess the accuracy of the relative SSH identification by comparing the magnitude of the resultant CS2 freeboards against drill hole measured freeboard. Additionally, we assessed linear and spatial trends in CS2 freeboard and CS2 ice thickness in this study. Any error in the relative sea surface height should not overly affect the trends because a constant median SSH value is subtracted from CS2 surface elevation measurements to obtain CS2 freeboards along each individual track. However, for automated freeboard retrievals and for regions of coastal sea ice without in situ measurements or open water nearby, poor identification of the relative SSH could introduce significant error and bias in the CS2 derived freeboard.'

Referee 1: Line 427-431: Please quantify these so that a comparison can be made

by specifying the increase you observed, using the SPL thickness measurements, to demonstrate how close the agreement is.

Author Comment: We have added the following statement to L426:

'The regional mean interpolated drill hole SPL thicknesses in the centre of McMurdo Sound was 3.90 m.'

Referee 1: Please include the magnitude of the higher freeboard in Price et al. (2013) and give an indication of the magnitude of the additional buoyant forcing of the sub-ice platelet layer.

Author Comment: Price et al. 2013 do not explicitly state the magnitude of the higher Multi-Year snow freeboard measured in the centre of McMurdo Sound. However, a maximum IceSat-1 measured freeboard of ~1.6 m is suggested in their Figure 10. Sea ice thickness measurements would be required to estimate the contribution of the SPL to this MY snow freeboard.

Referee 1: Technical Corrections Line 167: It would help to show individual tracks if they were plotted with slightly slimmer lines, especially for the East. Author Comment: Thank you for this comment, we will change the line thickness.

Referee 1: Figure 3: Missing label for x-axis Author Comment: Okay, thank you.

Referee 1: Line 355: Should 'd' be followed by a bracket ')'? Author Comment: Okay, thank you.

Additional References ARNDT, S., HOPPMANN, M., SCHMITHÜSEN, H., FRASER, A. D. & NICOLAUS, M. 2020. Seasonal and interannual variability of landfast sea ice in Atka Bay, Weddell Sea, Antarctica. The Cryosphere, 14, 2775-2793.

MAKSYM, T. & MARKUS, T. 2008. Antarctic sea ice thickness and snow‐to‐ice conversion from atmospheric reanalysis and passive microwave snow depth. Journal of Geophysical Research: Oceans, 113.

---

## Author Comment (AC2) · 26 Jan 2021

Authors Response to Anonymous Referee #2

Authors: We thank the reviewer for taking the time to review our manuscript. However, we do not agree with the reviewer's general assessment. We have considered each comment and modified the manuscript according to suggested changes where we agree and provided a justification where we do not. We hope that the responses given below have addressed the reviewer's comments.

Referee 2: Fast ice is an important component of the Antarctic climate system, especially for a better understanding of processes between ice shelf and ocean. It is therefore out of question that there is a great interest in the community to obtain better

[Figure]

estimates of the thickness of the solid ice over the whole area and thus to be able to conduct further calculations of sea ice volume in the next step.

Author Response: The objective of this study was not to obtain sea ice thickness to underpin sea ice volume calculations but to investigate whether a satellite altimeter (Cryosat-2 (CS2) available during our study period) can detect a known spatial pattern of higher freeboard driven by supercooled ISW in McMurdo Sound. It seems that the main objective of this study which was clearly identified by reviewers 1 and 3, and explicitly stated in the abstract, introduction and implicit throughout the text (L9-11; L14-15; L75-77; L85-87; L270-272; L324-327; L420-426; L434-435; L457-460; L465-472) has been overlooked by reviewer 2.

We will emphasise the main objective and highlight that this is the first study to apply satellite altimetry to specifically detect ice shelf-influenced fast ice freeboard by changing the statement on L85 to the following:

'For the first time, we investigate whether the CS2 satellite radar altimeter can detect the influence of ISW on fast ice in McMurdo Sound by consistently identifying the higher ice freeboard caused by thicker ice shelf-influenced fast ice combined with the buoyant forcing of the SPL beneath.'

Referee 2: However, the manuscript still shows fundamental weaknesses at this stage, so that it needs to be extensively revised before the work can be published. That is why I only mention here general remarks, which should be implemented first, before in a next iteration more detailed points can be raised.

Author Response: Please refer to the responses given the general comments below.

Referee 2: General Comments: Discussion paper 1. I see a major mismatch between the title, the objectives and the actual content of the manuscript. What is basically done in the manuscript is to compare measured on-site freeboard values related to fast-ice and platelet ice thickness with satellite retrieved freeboard values.

Author Response: This comment is not relevant if the reviewer has misunderstood the objective of the study as stated above "to obtain better estimates of the thickness of the solid ice over the whole area and thus to be able to conduct further calculations of sea ice volume in the next step."

The motivation is to use a standard satellite elevation product (ESA L2 Baseline C SIN product) and existing and proven methods to obtain fast ice freeboard (as developed and demonstrated in Price et al. 2015) to assess whether a satellite altimeter is capable of detecting a known pattern of higher freeboard driven by supercooled ISW outflow in McMurdo Sound.

We will emphasise the main objective and highlight that this is the first study to apply satellite altimetry to specifically detect ice shelf-influenced fast ice freeboard by changing the statement on L85 to the following:

'For the first time, we investigate whether the CS2 satellite radar altimeter can detect the influence of ISW on fast ice in McMurdo Sound by consistently identifying the higher ice freeboard caused by thicker ice shelf-influenced fast ice combined with the buoyant forcing of the SPL beneath.'

To do this, we obtained freeboard measurements from the CS2 satellite altimeter using a supervised freeboard retrieval procedure where the reference open water and relative Sea Surface Height was manually identified using satellite imagery (L215-220). That is, the CS2 freeboard was determined independently. The in situ freeboard measurements were used to assess and validate the satellite altimeter obtained freeboard and to constrain the best-match freeboard interface for ice thickness calculations (L223-224; L251-254). We then assessed regional spatial and linear trends in CS2 freeboard and CS2 ice thickness to discern if the CS2 radar altimeter could consistently detect the known pattern of fast ice freeboard (driven by the combination of the thicker ice shelf-influenced fast ice and the buoyancy of the SPL) in McMurdo Sound over multiple years as described throughout section 4.

Referee 2: However, this does not reflect the actual influence of ISW on the fast ice, as ISW is not necessarily the same as platelet ice.

Author Response: We do not understand this statement. 'Potentially' supercooled Ice Shelf Water (ISW) (i.e., the definition of ISW) in the upper surface ocean promotes sea ice formation and causes it to be thicker by stabilising the upper surface layer, by hindering vertical mixing and insulating sea ice from warmer subsurface waters below. Thicker Ice Shelf Water influenced sea ice inherently has a higher freeboard than sea ice without this influence.

Platelet ice is a direct manifestation and distinct signature of 'in situ' supercooled ISW at the ocean surface which causes the fast ice freeboard to stand higher through two effects: 1) the fast ice is thicker by platelet ice accumulation, and augmented growth through heat flux to the heat-deficit in the ocean induced by 'in situ' supercooled Ice Shelf Water, and 2) the buoyancy effect of the SPL if present.

Satellite altimetry measurements cannot differentiate between higher freeboard driven by 'in situ' or 'potentially' supercooled ISW but as showcased here, can identify a region where the fast ice freeboard near an ice shelf has significantly higher than average freeboard due to the influence of ISW in the upper surface ocean. The presence of 'in situ' supercooled ISW in the upper surface and its influence on fast ice and SPL formation in McMurdo Sound have been well observed in multiple studies as described in the introduction and explicitly stated on L87-89. When we refer to supercooled ISW in the text, we are referring to 'in situ' supercooling according to the commonly used nomenclature. To clarify, we will explicitly state this in the introduction.

Referee 2: 2. Therefore, a fundamental revision of clearly stated objectives is required, which are then also addressed accordingly in the manuscript.

Author Response: We reiterate that it seems that the main objective of this study which was clearly identified by reviewers 1 and 3, and explicitly stated in the abstract, introduction and implicit throughout the text (L9-11; L14-15; L75-77; L85-87; L270-272;

[Figure]

L324-327; L420-426; L434-435; L457-460; L465-472) has been overlooked by re-viewer 2.

Referee 2: 3. All applied methods in the manuscript are very poorly presented, so that it is not always 100% clear what was really done in detail. Therefore, in order to be able to judge the exact quality and reliability of the analyses presented, they must be made clearer. Furthermore, there is no clear distinction between the work/analyses presented here and what was done in previous work.

Author Comment: We thank the reviewer for their comment.

The motivation is to use a standard satellite elevation product (ESA L2 Baseline C SIN product) and existing and proven methods to obtain fast ice freeboard (as developed and demonstrated in Price et al. 2015) to assess whether a satellite altimeter is capable of detecting a known pattern of higher freeboard driven by supercooled ISW outflow in McMurdo Sound. To summarise, much of the method is not novel but the application is. We have made the following changes to emphasise the previous work by Price et al. 2014 and 2015 that underpins the methodology applied in this study.

1) To emphasise that the main method used in the study has been developed in detail in previous satellite altimetry work in McMurdo Sound, we have moved the paragraph in section 2 (L115-124) to L85 and added the following sentence at the end of this paragraph:

'Price et al. (2015) developed the method applied in this study to obtain CS2 fast ice freeboard in McMurdo Sound in 2011 and 2013 and the relevance of this work to this study is described in more detail in section 2.2'

We will emphasise that this is the first study applying satellite altimetry to specifically detect ice shelf-influenced fast ice freeboard by changing the statement on L85 to the following:

'For the first time, we investigate whether the CS2 satellite radar altimeter can detect

the influence of ISW on fast ice in McMurdo Sound by consistently identifying the higher ice freeboard caused by thicker ice shelf-influenced fast ice combined with the buoyant forcing of the SPL beneath.'

We will also highlight and emphasise the methods that are novel throughout the text. Novel aspects of the study are as follows:

- The application of satellite altimetry to identify a known pattern of higher freeboard caused by thicker ice shelf-influenced fast ice combined with the buoyant forcing of the SPL which both result from supercooled ISW outflow.

- The method to identify the best-matching freeboard interface for individual CS2 tracks as described in L251-260 and L401-405.

- Calculation of ice thickness from CS2 freeboard in McMurdo Sound for multiple years and comparison with interpolated in situ measured ice shelf-influenced fast ice and SPL and their combined Mass Equivalent Thickness (MET). L236-249; L290-327; Figures 4 and 5.

- Assessment and comparison of regional trends in CS2 freeboard (Table 1), and linear trends in CS2 ice thickness towards the McMurdo Ice Shelf in a region with significant ISW influence (centre) and another region with less pronounced ISW influence (east) (Figure 4).

- Assessment of spatial patterns of CS2 freeboard and CS2 ice thickness and comparison with in situ observed distributions of ice shelf-influenced fast ice and SPL (presented as a combined Mass Equivalent Thickness (MET)) (Figure 5).

Referee 2: 4. The snow cover on Antarctic sea ice is known to play a crucial role in both remote sensing and the buoyancy principle of sea ice. Even though the time series shown here seems to have a negligible snow thickness, this cannot be neglected in such a study. Instead, much more attention must be paid to potential difficulties caused by superimposed ice, snow ice or severe snow metamorphism.

Author Response: The snow was not neglected in our analyses by any means. In fact, this study is rare in that we had four years of in situ measurements of snow depth available to be able to constrain the effects of the snow layer. The importance of snow is dealt with throughout the document, and the limitation is revisited in the outlook section on L438 and L475 of the conclusion. Snow depressed negative freeboard or surface flooding was not observed, and we have added the following sentence on line 152 to clarify this:

"The snow layer can depress the freeboard and result in flooding of the sea ice surface and the formation of meteoric ice which can contribute to freeboard (Maksym and Markus, 2008). Snow-depressed negative freeboard or surface flooding were not observed at drill hole sites in McMurdo Sound in late spring. Multiple ice core studies carried out in the region over winter and in late spring revealed no contribution of meteoric ice to the fast ice cover in McMurdo Sound (e.g. Dempsey et al., 2010, Gough et al., 2012)'

We discuss how the snow layer in McMurdo Sound would affect the results of this study on L412-417 highlighting that the snow distribution from west to east was advantageous in that we could have confidence in that the trends in higher freeboard in the centre of the sound (where the snow is thin and loosely-packed) did not result from the addition of the snow layer which would have a more significant effect in the east where the snow is deeper.

Referee 2: 5. Moreover, this very thin layer of snow raises serious doubts as to the extent to which such a study can be used beyond the study area shown here. Indeed, a stronger positive freeboard of fast ice areas is not only due to the buoyant platelet layer. Instead, studies, e.g. in Atka Bay (Arndt et al., 2020), have shown that it is not the platelet ice that is the decisive component for the freeboard of the fast ice, but the snow cover. I therefore strongly recommend to do similar sensitivity studies for this region - and also to use data from previous years to emphasize that in McMurdo Sound there is always this low snow load. Even if this is the case, the conclusions must still

be strongly weakened, because a strong positive freeboard can have different reasons – which cannot be quantified with CryoSat alone.

Author Response: McMurdo Sound does present favourable conditions for such a study with relatively thin snow cover and smooth fast ice (the potential limitation of inadequate snow knowledge is reiterated on L438 and in the conclusion on L475). This, along with the availability of multiple years of in situ measurements and detailed knowledge of ISW processes in the region is why we chose McMurdo Sound for our analysis as stated on L87-89. This study is a proof of concept demonstration (L486-488)) that CS2 is capable of detecting this signal of ISW influence in fast ice freeboard with the right conditions and information. It is a first step in the development of a satellite-based method (L434) to identify and potentially monitor regions where ISW (in situ supercooled or not) is outflowing in the upper surface ocean and is influencing fast ice formation.

The community has very little knowledge of the snow distribution around Antarctica and we will not know if this can work in other locations until we try. However, this proof-of concept study shows that it can work in the conditions in McMurdo Sound and is certainly worth pursuing elsewhere, especially when combined with auxiliary satellite information. ICESat-2 data is now available and is providing highly detailed snow freeboard information which has promise to be used in tandem with CS2 to obtain information on the snow cover as stated on L439-441.

We are aware of the Arndt et al. 2020 study and the frequent negative freeboard they observed from snow loading given that the snow depths in Atka Bay were up to 0.89 m. Atka Bay and McMurdo Sound represent a tiny fraction of a ~45, 000 km Antarctic coastline, 74% of which is fringed by outlet glaciers and ice shelves. This in itself highlights the paucity of observations of ice shelf-influenced fast ice or indeed ordinary fast ice on the Antarctic coastline and the dependence of our knowledge of fast ice/ISW processes on a small handful of sampling locations that happen to be situated near and accessible from Antarctic research bases. There is very much a need for a satellite-
based method to identify these important regions. We will reiterate throughout the text that this is a proof of concept study and further work is required to reveal similar areas of ISW influenced fast ice elsewhere on the Antarctic coastline. To acknowledge this limitation, we have added the following statement in section 5.4 on L435:

'The smooth gradients in fast ice and SPL thickness and low snow coverage in Mc-Murdo Sound present favourable conditions for the CS2 radar altimeter to detect higher ice shelf-influenced freeboard. However, more challenging conditions for satellite altimetry are likely to be presented elsewhere on the Antarctic coastline. A recent drill hole assessment of supercooled ISW-influenced fast ice in Atka Bay observed deep snow accumulations of up to 0.89 m which resulted in frequent negative fast ice freeboard regardless of the buoyant forcing of a substantial SPL beneath (Arndt et al., 2020). As far as we are aware, Atka Bay and McMurdo Sound are the only two locations on the Antarctic coastline with multiple years of in situ measurements of ice shelf-influenced fast ice, SPL and snow highlighting the need for a satellite-based method to identify other regions where ISW is present in the upper surface ocean and influencing fast ice formation.'

Referee 2: 6. Referring to the previous point, the work shown here would greatly benefit from putting the measured snow, fast ice and platelet ice thicknesses into a more global context with measurements from other regions or other points in time in the same region.

Author Response: As far as we are aware McMurdo Sound and Atka Bay are the only locations where this level of detailed in situ information on coincident snow depths, fast ice and sub-ice platelet layer thickness has been collected and published. We are confident that these are the only locations where multiple years of these measurements have been collected. We have now referred to the Arndt study to highlight the challenge of the snow in the successful application of this method in L435 as stated in the previous comment. We do not understand how this work can be put into a 'global context' when this process has only been observed to occur in Antarctica, and in this detail

in only two locations in Antarctica. The 'other points in time in the same region' was addressed by applying our assessment over four years and then discussing the findings extensively in the text with respect to previous work carried by Brett et al. (2020), and Price et al., (2014), (2015), (2019) amongst others.

Referee 2: Otherwise, the results found here are unfortunately not very reliable and raise great doubts that they can be applied on a larger scale.

Author Response: We question how the results presented in this study are not very reliable and strongly disagree with this comment. In fact, we present arguably some of the most reliable CS2 information obtained in the Antarctic as it has been very well validated with multiple years of in situ information in a region with well understood ice formation and ISW processes. We would like to reiterate that this study is rare in that we had multiple years of in situ measurements of coincident freeboard, fast ice and sub-ice platelet layer thickness and snow depths to validate and compare with the independently obtained CS2 freeboard. The only other location with this information available (and published) is in Atka Bay which as the reviewer has already pointed out is subject to deep snow accumulations and thus significant snow loading and depression of the fast ice freeboard. As explained in the manuscript, the applicability of our method to a larger scale will critically depend on the quantification of the snow layer (L438; L449-451).

Additional References ARNDT, S., HOPPMANN, M., SCHMITHÜSEN, H., FRASER, A. D. & NICOLAUS, M. 2020. Seasonal and interannual variability of landfast sea ice in Atka Bay, Weddell Sea, Antarctica. The Cryosphere, 14, 2775-2793.

MAKSYM, T. & MARKUS, T. 2008. Antarctic sea ice thickness and snow‐to‐ice conversion from atmospheric reanalysis and passive microwave snow depth. Journal of Geophysical Research: Oceans, 113.

---

## Author Comment (AC3) · 26 Jan 2021

Author Response to Anonymous Referee #3

Authors: We thank the referee for taking the time to review our manuscript and for providing valuable feedback. We have considered your comments and modified the manuscript according to suggested changes where we agree and provided a justification where we do not. We hope that the responses given below and modifications made have addressed the reviewer's comments.

General Comments.

Referee 3: The paper examines the ability of CryoSAT 2 to detect the existence of a sub-ice platelet layer found under McMurdo Sound fast ice and previously determined

to result from the upwelling of supercooled Ice Shelf Water. The study provided good comparisons with the freeboard rise from satellite altimetry with a detailed ground truth campaign conducted over four years of measured sea ice freeboard, snow depth, sea ice thickness and sub-ice platelet layer thickness distributions. Proof of the utility of satellite altimetry to effectively determine the distribution of these sub-ice platelet layers even in a selected region like McMurdo Sound is worthy of publication given that the lack of ground truth in other sea ice studies is a continuous impediment to progress in maximizing the potential of satellite remote sensing to effective monitoring of sea ice processes from space. There are some needed revisions to fully realize the paper's potential.

Specific Comments.

Referee 3: Proof of the utility of satellite altimetry to determine the distribution of these sub-ice platelet layers will provide an effective means of monitoring them from space and help in monitoring the interannual variability of the flux of Ice Shelf Water from underneath the Ross and McMurdo Ice Shelves in future.

That said, there are some difficulties in the presentation of the results. For example, in Figure 5, the use of the same color bars for quite different scales, CS2 freeboard in 5a (up to 0.6m), CS2 ice thickness in 5b (up to 5.8m) and drill hole MET in fig 5c (up to 3.5m) is difficult to interpret correctly. (note also in Technical Comments about the need for intermediate values).

Author Response: We thank the reviewer for this comment. We will use different colour bars for the three parameters.

Referee 3: In the abstract the sentence "We demonstrate the capability of CryoSat-2 to detect higher Ice Shelf Water influenced fast ice freeboard in McMurdo Sound and the wider application of this method as a potential tool to identify regions of ice shelf-influenced fast ice elsewhere on the Antarctic coastline." Is a reach too far, given the unique condition of McMurdo with its generally very thin snow cover which may not be

generally found in other coastal regions. There is also no attempt in the paper itself to apply the technique to other regions. Suggest limiting the statement to only: "We demonstrate the capability of CryoSat-2 to detect higher Ice Shelf Water influenced fast ice freeboard in McMurdo Sound." In the discussion of the paper, the concepts can be best given as to applicability to other regions, with sufficient caveats given as to the role of thicker snow than found in McMurdo Sound for example, and how this may affect the interpretations elsewhere.

Author Response: We thank the reviewer for this comment and agree that McMurdo sound presents favourable conditions. We meant to say in this statement that it could work elsewhere and have now clarified this as below:

"We demonstrate the capability of CryoSat-2 to detect higher Ice Shelf Water influenced fast ice freeboard in McMurdo Sound. Further development of this method could provide a tool to identify regions of ice shelf-influenced fast ice elsewhere on the Antarctic coastline with adequate information on the snow layer."

Technical Comments.

Referee 3: Abstract: "We attribute this overestimate in satellite altimeter obtained ice thickness to the additional buoyant forcing of the sub-ice platelet layer. " Comment: Need to know if the measurement of the sub-ice platelet layer distribution verified this.

Author Response: Yes, this was obtained from the central region of McMurdo Sound where the SPL is consistently thicker every year. We have added the following statement to the abstract and L426:

'The regional mean interpolated drill hole SPL thicknesses in the centre of McMurdo Sound was 3.90 m.'

Referee 3: Line 124: What were the equivalent freeboard rises in cm to the 12% and 19% freeboard increases at those locations? Were there also thickness measurements of the SPL at that time?

Author Response: The 12 % is the mean deviation in calculated sea ice thickness for the fast ice in McMurdo Sound and 19 % the maximum deviation. Price et al., 2014 do not state the equivalent rises in freeboard or for these overestimated sea ice thickness values and do not indicate the location where the maximum deviation of 19 % was observed. Price et al 2014 did assess in situ SPL measurements in their analysis.

Referee 3: Line 139 (Grammar error)."Refer to Brett et al. (2020) for a detailed description of the thickness distributions of ice shelf-influenced fast ice, the SPL and snow in McMurdo Sound in November of 2011, 2013 and 2017." Change to: Brett et al. (2020) provide a detailed description of the thickness distributions of ice shelf-influenced fast ice, the SPL and snow in McMurdo Sound in November of 2011, 2013 and 2017.

Author Response: Thank you, we will change the sentence according to this suggestion.

Referee 3: Line 140 Change to: Here we summarize those descriptions to show general patterns and also include the fast ice conditions in 2018.

Author Response: Okay, thank you.

Referee 3: Line 146 give a value for more substantial deposition of snow

Author Response: Thank you. We have included a range of magnitudes observed for snow depth in the east and southeast as ~0.2-0.4 m

Referee 3: Figure 1. Give some intermediate SPL values on the color bar rather than just the High and Low. Might also include a few (4 or 5) of these as identified contour lines on the map plot. In the caption point out that the red square on the inset map is the area (McMurdo Sound) shown on the MODIS image to the right.

Author Response: Thank you, we will provide intermediate values following this suggestion.

Referee 3: Figure 5 need intermediate values on the color bar (Freeboard, C2-2 Ice

thickness, Drill Hole MET) and address concerns about the same color bars but different scales for a, b and c.

Author Response: We will add intermediate values and use different colour bars for the three parameters.

Referee 3: Line 490 Appendix A Seems hard to read,

Author Response: Apologies, we are uncertain what you refer to here. The processing applied to the CS2 product is very technical and could easily dominate this study. Here, we aimed to build on previous assessments of CryoSat-2 by others (e.g., Price et al., 2015) to focus on our objective of using satellite altimetry to detect an ice shelf/fast ice process. In Appendix A, we are ensuring that the geophysical corrections and the geoid detrending are robust and produce along-track CS2 surface elevation profiles with minimal residual curvature remaining from these geophysical effects over open water in late summer, and expected fast ice freeboard magnitudes and trends in late spring. As commented below, we will better clarify the applied corrections for each surface type and we will make the text easier to read throughout Appendix A.

Referee 3: Line 530 This indicates that the geoid is de-trended for twice (?? Don't understand this sentence)

Author Response: The Mean Sea Surface (MSS) is a combination of the geoid surface and Mean Dynamic Topography. Therefore, applying the MSS correction should account for the geoid surface. However, we observed that the geoid is not accounted for in the 7 years of CS2 measurements of the open ocean surface (in the shape and magnitude of the CS2 height). We would expect surface elevation retrievals of an approximate magnitude of -2 m relative to WGS84 ellipsoid (Figure A1b) when the geoid has been detrended and not -55 m (Figure A1a). The residual curvature in the shape of the surface was very similar to the EGM 2008 geoid and when we detrended for the EGM 2008 modelled geoid we obtained the desired flat profiles. We additionally assessed this phenomenon in late spring fast ice conditions and the resultant detrended

profiles aligned with in situ measured freeboard. We explored and read extensively about the CS2 data product and associated corrections but could find no solution or explanation to this phenomenon. As stated in the text the information provided by ESA was very limited and we were unable to clarify what MSS model was being applied in our study region. . . . . . We suspect it may be a processing error but we could not confirm this. We will clarify in the text that this is a potential processing error in the product and rewrite L528-531 according to information given in this response.

Referee 3: Lines 516-532 (Appendix A). I find this discussion rather confusing, perhaps hampered by my own limited knowledge of Geodesy. For example, the sentence (Line 507) "The open ocean surface with MSS and ocean/tidal corrections removed was consistent from 2011 to 2017." Is this both MSS and ocean/tidal corrections removed (Item 4) or Is it Item 3, MSS applied and ocean /tidal corrections removed? I infer that the authors' find the best practice for surface elevation is Item 6. in their first list but this is difficult to suss out from their discussion. Suggest a Table listing the various options and some index of performance e.g. Good, Fair , Poor along with better referencing of the number of the option in the accompanying text would help to clarify.

Author Response: We thank the reviewer for this comment and we will better clarify the applied corrections for each surface type and make the text easier to read throughout Appendix A.

Referee 3: Line 542 "The in situ data included in this study will not be available at the time of publication but it is intended that it will be deposited in a data repository." However from the Journal Data Policy:"Copernicus Publications requests depositing data that correspond to journal articles in reliable (public) data repositories, assigning digital object identifiers, and properly citing data sets as individual contributions: : :.. Authors are required to provide a statement on how their underlying research data can be accessed. This must be placed as the section "Data availability" at the end of the manuscript."

Referee 3: Reviewer Comment: Further revisions should include the citation to Data Availability required by this journal. A substantive further review may require the reviewers' and editor to examine the data used in the paper before final approval and the paper cannot be examined by others without the ability to further examine the data and conduct their own analyses.

Author Response: All CS2 data used in this study is available as stated in the 'Data Availability' section and this can be used to replicate most of the analyses. The in situ data will be made available and the following data availability statement will be included on L542.

'The remaining data will be made available at the World Data Center PANGAEA https://www.pangaea.de.'

---

## Author Response (AR1)

Dear Editor Dr Melody Sandells,

Thank you very much for managing our manuscript and for your decision to publish the manuscript with revisions.

As directed, we have made the changes set out in the Editor's report and all the proposed modifications we listed in the responses to Reviewers 1, 2 and 3. We have provided a revised version (Revision 1) and a tracked changes (TRACKED) version with for your consideration. We have not included a point by point account here of all changes made as we would be repeating the information provided in the responses to reviewers. Please let us know if this causes any inconvenience and we will provide a list of all changes made.

In response to the Editors comments, we have:

- Emphasised that the main methodology builds on previous work of Price et al., 2014 and 2015.
- Referred to SAR Interferometry as InSAR throughout the text.
- Added a description of the freeboard signature of fast ice under the influence of 'potential' and 'in situ' supercooled ISW to Section 5.4 on L489-497 of the most recent revised version.
- Added a list of novel aspects of this study to Section 5.3 of the discussion on L461-471 of the most recent revised version.

Thank you very much again for further consideration of our manuscript. We look forward to working with you for the final iteration.

Best wishes,

Gemma Brett (on behalf of all co-authors).

---

## Referee Report (RR1)

**Review on "Satellite altimetry detection of ice shelf-influenced fast ice" by Gemma M. Brett et al.**

I appreciate the comprehensive work of the authors addressing the expectations of all reviewer comments. I do therefore see an essential improvement of the manuscript, especially regarding the clarification of the objectives and the subsequent discussion of the gained knowledge. Thus, the manuscript is from my perspective ready for publication.

---

## Editor Decision (ED1)

Thank you for addressing the comments of the reviewers and making substantial changes, particularly to the discussion section of the paper. I would hope that these changes meet most of the reviewers' expectations. There are still a few issues that need further attention. Please make the following changes to progress the paper:

Two of the reviewers requested greater clarity of the novel aspects of this study. Although the response to the reviewers indicated that the method is not new the application is (which is absolutely fine), this conflicts with the statement on line 464 that 'The method to identify the best-matching freeboard interface for individual CS2 tracks (sect. 3.1).' is novel and on line 282 'We applied a novel technique to select the best-matching freeboard interface for each track'.

It is essential to be clear on how this study differs from Price et al. (2019). Section 3.1 contains the equations already presented in Price et al. (2019) but even if this study uses the method of Price et al. (2019) the level of detail on the best-matching freeboard method either in this paper or Price et al. (2019) is insufficient to allow this study to be repeatable (e.g. what constitutes a 'match' / what cost function was used to find the 'penetration depth')? If development from Price et al. (2019) is to use equations 1 and 2 first, then resort to equation 3, this is not particularly new as equation 3 is the same as equation 1 if Pd=0 or equation 2 if Pd=Ts. As a minimum the equations must be included, but it would be far better to provide the code to complete the analysis (as per Data Policy for The Cryosphere) with the publication. In addition, I would avoid the terminology 'penetration depth' as this is not a measure of 1/e reduction in electromagnetic radiation.

More information on the data used under Data Availability is welcomed, but please refer to the specific DOI for the in situ data and cite them (please see https://www.the-cryosphere.net/policies/data_policy.html) to give appropriate credit to the groups who collected these data.

Please revisit the colour scheme for Figure 5. Fig 5b and 5c have a similar colour bar but opposite in direction, which is confusing. Please use divergent colour schemes (Fig 5a is good) rather than rainbow schemes, which are hard to interpret for those who are colour-blind. I recommend checking all images with an online colour-blindness checker.

---

## Author Response (AR2)

Dear Editor Dr Melody Sandells and Reviewers,

Thank you very much for taking the time to review our manuscript once again and for providing valuable feedback. We really appreciate your efforts which have much improved the manuscript. In response to your comments, we provide clarification and description of revisions below. We have included a revised version (Revision-2) and a tracked changes version (TRACKED-2) for your consideration. Line numbers refer to the PDF document 'TRACKED-2_Brett-et-al_Satellite-altimetry-detection-of-ice-shelf-influenced-fast-ice'.

Thank you very much for reviewing our manuscript.

Best wishes,

Gemma Brett (on behalf of all co-authors).

**Editor Comment:**

Thank you for addressing the comments of the reviewers and making substantial changes, particularly to the discussion section of the paper. I would hope that these changes meet most of the reviewers' expectations. There are still a few issues that need further attention. Please make the following changes to progress the paper:

Two of the reviewers requested greater clarity of the novel aspects of this study. Although the response to the reviewers indicated that the method is not new the application is (which is absolutely fine), this conflicts with the statement on line 464 that 'The method to identify the best-matching freeboard interface for individual CS2 tracks (sect. 3.1).' is novel and on line 282 'We applied a novel technique to select the best-matching freeboard interface for each track'. It is essential to be clear on how this study differs from Price et al. (2019). Section 3.1 contains the equations already presented in Price et al. (2019) but even if this study uses the method of Price et al. (2019) the level of detail on the best-matching freeboard method either in this paper or Price et al. (2019) is insufficient to allow this study to be repeatable (e.g. what constitutes a 'match' / what cost function was used to find the 'penetration depth')? If development from Price et al. (2019) is to use equations 1 and 2 first, then resort to equation 3, this is not particularly new as equation 3 is the same as equation 1 if Pd=0 or equation 2 if Pd=Ts. As a minimum the equations must be included, but it would be far better to provide the code to complete the analysis (as per Data Policy for The Cryosphere) with the publication. In addition, I would avoid the terminology 'penetration depth' as this is not a measure of 1/e reduction in electromagnetic radiation.

**Author Response:**
We thank the editor and the reviewers for this comment. Our apologies, we agree that the principle of the method is much the same as that applied in Price et al., 2019. We have thus removed all statements that this method is novel and have also changed 'penetration depth' to 'penetration factor'.

Price et al., 2019 explored the range of ice thicknesses obtained from CS2 freeboard when using different snow products and hydrostatic equations. They then assessed ice thicknesses obtained from CS2 freeboard using interpolated in situ measured snow depth and a range of penetration factors (refer to their Figure 6). They compared the mean interpolated in situ

measured Mass Equivalent Thickness (MET) (i.e., what CryoSat-2 measures) with the mean CS2 ice thicknesses obtained from a range of penetration factors. They found the closest agreement of mean CS2 ice thickness and in situ measured MET when the penetration factor was 0.07 m (shown in their Figure 6). Those mean values were calculated over a defined fast ice area where the in situ measurements were made.

In this study, we compared the profiles of calculated CS2 ice thickness and drill hole measured Mass Equivalent Thickness along each individual track. To select the best matching freeboard interface and corresponding hydrostatic equation, we assessed 1) the along-track profiles of interpolated in situ MET with the CS2 ice thicknesses visually (this was evident if the best interface was sea ice or snow freeboard), 2) linear regression analyses if there was a gradient in thicknesses, or 3) by comparing mean values along-track according to Price et al., 2019. We tried to identify the optimum freeboard interface using the in situ and CryoSat-2 information, as opposed to assuming some arbitrary or constant freeboard interface which would have simplified the analyses but also weakened the results. The main result (shown in Figures 5 and 6) is that the CryoSat-2 ice freeboard and ice thickness in the centre of McMurdo Sound increased towards the ice shelf with increasing in situ measured ice shelf-influenced fast ice and SPL. In the east, the gradients were flat because the influence of supercooled ISW is less pronounced and CryoSat-2 detected this. The snow distribution was also advantageous as we could be assured that the higher CS2 freeboard and increasing trends in the centre were not due to the snow cover which was generally thin and loosely packed (Line 452-453; and 463-469). We have better clarified this in the text on Line 287-294, referred to Price et al., 2019 (Line 218 and 294) and discussed potential limitations on Line 444-447

**Editor Comment:**
More information on the data used under Data Availability is welcomed, but please refer to the specific DOI for the in situ data and cite them (please see https://www.the-cryosphere.net/policies/data_policy.html) to give appropriate credit to the groups who collected these data.

**Author Response:**
Our apologies, the specific DOIs for the individual 2011, 2013 and 2017 in situ datasets were not available at the DOI provided in Haas et al., 2020. This data submission has been complicated by the datasets being collected under different projects, funding, principal investigators and field teams and being supplement to multiple publications since 2009. As a co-author on that publication, I have taken over managing this data submission on behalf of all co-authors and can assure you that the DOIS for all data will be available soon.

The 2018 dataset has been submitted and is under review in Pangaea. We are still waiting for a temporary DOI and have been informed by the Pangaea editorial team that it could take several months for the data to be reviewed and the DOI to be fully minted.

**Editor Comment:**
Please revisit the colour scheme for Figure 5. Fig 5b and 5c have a similar colour bar but opposite in direction, which is confusing. Please use divergent colour schemes (Fig 5a is good)

rather than rainbow schemes, which are hard to interpret for those who are colour-blind. I recommend checking all images with an online colour-blindness checker.

**Author Response:**
Thank you for this comment. We have taken this into consideration and changed the colour schemes of Figures 1 and 5b and 5c to divergent colour schemes.

---

## Author Response (AR3)

Dear Melody and reviewers,

Thanks very much for the final review and acceptance of our paper.

We have made the technical corrections as directed. My co-authors have brought to my attention that the CryoSat-2 combined SAR and interferometer altimetric mode is referred to as SARin or SIN according to the ESA nomenclature (https://earth.esa.int/eogateway/instruments/siral/description/) and not InSAR (which was my error in a previous revision) which refers to imaging radar.

Apologies for the confusion.

Many thanks again. It has been a pleasure to work with you all.

Best wishes,

Gemma Brett (on behalf of all co-authors).